

# The impact of regional-scale upper mantle heterogeneity on glacial isostatic adjustment in West Antarctica

Erica M. Lucas[1], Natalya Gomez[1], Terry Wilson[2]

[1] Earth and Planetary Sciences Department, McGill University, Montreal, Canada
[2] School of Earth Sciences, Ohio State University, Columbus, OH, United States of America

*Correspondence to*: erica.lucas@mcgill.ca

**Abstract.** West Antarctica is underlain by a laterally heterogenous upper mantle, with localized regions of mantle viscosity reaching several orders of magnitude below the global average. Accounting for 3-D viscosity variability in glacial isostatic adjustment (GIA) simulations has been shown to impact the predicted spatial rates and patterns of crustal deformation, geoid, and sea level changes in response to surface ice loading changes. Uncertainty in the viscoelastic structure of the solid Earth remains a major limitation in GIA modeling. To date, investigations of the impact of 3-D Earth structure on GIA have adopted solid Earth viscoelastic models based on global- and continental-scale seismic imaging, with variability at spatial length scales > 150 km. However, regional body-wave tomography shows mantle structure variability at smaller length scales (~50-100 km) in central West Antarctica. Here, we investigate the effects of incorporating smaller-scale lateral variability in upper mantle viscosity into 3-D GIA simulations. Lateral variability in upper mantle structure at the glacial drainage basin scale is found to impact GIA model predictions for modern and projected ice mass changes, especially in coastal regions that undergo rapid ice mass loss. Differences between simulations adopting upper mantle viscosity structure inferred from regional- versus coarser continental-scale seismic imaging are large enough to impact the interpretation of crustal motion observations and reach up to ~15% of the total predicted sea level change during the instrumental record. Incorporating a strong transition from lower viscosities at the mouth of the Thwaites and Pine Island glaciers to higher viscosities in the interior of the glacier basins results in a ~10-20% difference in predicted sea level change in the vicinity of the grounding line over the next ~300 years. These findings have a range of implications for the interpretation of geophysical observables and improving constraints on feedbacks between the West Antarctic Ice Sheet and the solid Earth.





## 1 Introduction

Of the West Antarctic Ice Sheet, the Amundsen Sea Embayment sector (ASE) (Fig. 1a) is the

dominant contributor to sea level rise at the present and will likely remain a primary contributor

for decades to come (e.g., Rignot et al., 2019; Shepherd et al., 2019; DeConto et al., 2021; Seroussi

et al., 2023). Within the ASE, the Pine Island and Thwaites glacier drainage basins, which

encompass the Pope, Smith, and Kohler glaciers, have undergone the greatest ice mass losses in

recent decades (Shepherd et al., 2019). Along with a wide range of cryospheric and climatic

processes, glacial isostatic adjustment (GIA) - the deformational and gravitational response of the

solid Earth to changes in ice mass distribution - must be accounted for when evaluating the current

and future stability of Thwaites Glacier (TG), Pine Island Glacier (PIG), and the West Antarctic

Ice Sheet more broadly (e.g., Gomez et al., 2015; Konrad et al., 2015; Barletta et al., 2018;

Kachuck et al., 2020; Coulon et al., 2021; Book et al., 2022; Gomez et al., 2024). GIA also

introduces substantial uncertainty into modern ice mass loss estimates derived from satellite

remote sensing (King et al., 2012; the IMBIE team, 2018; Valencic et al., 2024). Improving the

accuracy of GIA predictions is not only important for advancing our current understanding of ice

sheet stability but is critical for the interpretation of geophysical and geological records of ice

change.


Accurate predictions of GIA are heavily reliant upon estimates of mantle viscosity and lithospheric

thickness in Antarctica (e.g., Barletta et al., 2018; Nield et al., 2018; Powell et al., 2020, 2022;

Wan et al., 2022). While many studies have adopted 1-D varying Earth structure when modeling

GIA in Antarctica, recent work has demonstrated the importance of accounting for realistic 3-D

Earth structure in GIA models (e.g., Powell et al., 2020; Wan et al., 2022; Gomez et al., 2024). To

develop models of the Earth's viscosity structure for input to 3-D GIA models, studies typically

infer viscosity from seismic wave speeds and other physical parameters (e.g., Ivins & Sammis,

1995; Latychev et al., 2005a; Paulson et al., 2005; Wu, 2005; Ivins et al., 2023). Improved seismic

station coverage across Antarctica over the past two decades has permitted continental-, regional-

, and local-scale seismic investigations of upper mantle structure from the analysis of passive

seismic data (e.g., Wiens et al., 2023). Investigations spanning a range of geographic scales have

revealed strong lateral variations in upper mantle seismic structure across West Antarctica (e.g.,

Hansen et al., 2014; Lloyd et al., 2015; Heeszel et al., 2016; Shen et al., 2018; O'Donnell et al.,



2017, 2019; White-Gaynor et al., 2019; Lloyd et al., 2015, 2020; Lucas et al., 2020, 2021, 2022).
Upper mantle velocities near the global average reference velocity are found across the interior of
West Antarctica (e.g., Lloyd et al., 2020). Low upper mantle velocity anomalies have been imaged
beneath Marie Byrd Land extending along the ASE coast, beneath the mouths of TG and PIG, and
towards the Antarctica Peninsula (e.g., Heeszel et al., 2016; Shen et al., 2018; O'Donnell et al.,
2019; Lloyd et al., 2020; Lucas et al., 2020). Localized low velocity anomalies have also been
imaged in the interior of central West Antarctica in regional-scale seismic investigations (e.g.,
Lloyd et al., 2015; Lucas et al., 2020). Here we define central West Antarctica as the region
approximately outlined in Fig. 1e, which encompasses the Thwaites and Pine Island glacier
drainage basis and extends west into Marie Byrd Land. As referred to here, the Amundsen Sea
Embayment sector (ASE) is the region of central West Antarctica that sits along and adjacent to
the coast (Fig. 1).

While previous 3-D GIA modeling studies have adopted mantle viscosity structure from
continental-scale seismic imaging, glacial-basin scale investigations of GIA have remained elusive
due to limited seismic resolution. However, benefiting from improved seismic station coverage in
West Antarctica, recent regional- and local-scale seismic imaging has revealed notable
heterogeneity in upper mantle seismic velocities within the TG and PIG glacial drainage basins
("glacial drainage basins" will hereafter be referred to as "basins" for simplicity) (Lucas et al.,
2020, 2021). The heterogeneity observed within the TG and PIG basins is consistent with the
presence of upper mantle viscosity changes up to ~2 orders of magnitude over geographic length-
scales of ~50-200 km.

In this study, we evaluate the impact of regional-scale variability in upper mantle viscosity on
predictions of changes in relative sea level (i.e. the height of the sea surface equipotential relative
to the solid surface), crustal deformation, and geoid in response to modern (last ~25-125 years)
and projected (next ~300 years) ice mass changes in West Antarctica. For modern ice mass
changes, we explore the impact of short-wavelength upper mantle viscosity variations on the
interpretation of crustal motion rates observed at GPS sites across central West Antarctica. We
also assess how regional-scale upper mantle viscosity variations impact bedrock elevation



predictions at the grounding line of TG and PIG and discuss the implications of our findings for
placing improved constraints on solid Earth – ice sheet – sea level feedback processes.

## 2 Methods

To investigate the length-scale over which lateral variations in upper mantle viscosity impact GIA
predictions, we perform a suite of simulations with Earth models that incorporate upper mantle
viscosity variations in West Antarctica inferred from regional- and continental-scale seismic
imaging. We adopt the Seakon model, a global, 3-D, finite-volume GIA forward model (Latychev
et al., 2005a), with regional grid refinement (as described in Gomez et al., 2018) for all simulations.
The Seakon GIA model solves the sea-level equation (Kendall et al., 2005) with time-varying
shorelines and computes the response of an elastically compressible Maxwell viscoelastic Earth to
a specified ice loading history while accounting for Earth rotational effects. Various surface
observables, including vertical and horizontal crustal displacements, relative sea level, and geoid
changes, are predicted by Seakon. Computations are performed on a global tetrahedral grid
comprised of ~28 million grid nodes and ~160 million elements. The adopted computational grid
is regionally refined in our study area, with a lateral surface resolution of ~3 km over central West
Antarctica, ~7 km over the rest of Antarctica, and ~12-15 km globally outside of Antarctica.
According to the Wan et al. (2022) sensitivity analysis of grid resolution on predictions of GIA,
the computational grid adopted here is adequate for capturing the GIA response resulting from
modern and future ice mass loading changes. Lateral resolution in the computational grid decreases
with depth, with coarsest resolution of ~50 km at the core-mantle boundary. Seakon's grid
refinement capabilities are advantageous for this study as they permit the incorporation of short-
wavelength lateral variability in solid Earth structure. We describe the Earth structure and ice cover
model inputs to Seakon below.

### 2.1 Earth model

To test the influence of incorporating various degrees of lateral upper mantle heterogeneity on GIA
predictions, we adopt one 1-D (i.e., radially varying) Earth model and three 3-D viscoelastic Earth
models with variations in upper mantle viscosity inferred from global-, continental-, and regional-
scale seismic velocity models. For all Earth models, lateral variations in lithospheric thickness are
based on the LithoRef18 global model (Afonso et al., 2019) combined with the model of Wiens et



al. (2023) in Antarctica (Fig. 1b). As in Wan et al. (2022) and Gomez et al. (2024), the elastic and density structure for all Earth models varies radially and is based on the STW105 seismic tomography model (Kustowski et al., 2008). The 1-D Earth model, which has an upper mantle viscosity of $10^{19}$ Pa s (bottom of lithosphere to 670 km depth) and a lower mantle viscosity of 5 x $10^{21}$ Pa s (670 km to core-mantle boundary), is calibrated to best reflect absolute upper-mantle

viscosity estimates from GIA models with the best fits to GPS uplift rates in West Antarctica (Nield et al., 2014; Zhao et al., 2017; Barletta et al., 2018; Samrat et al., 2021). The 1-D Earth model will hereafter be referred to as "1D".

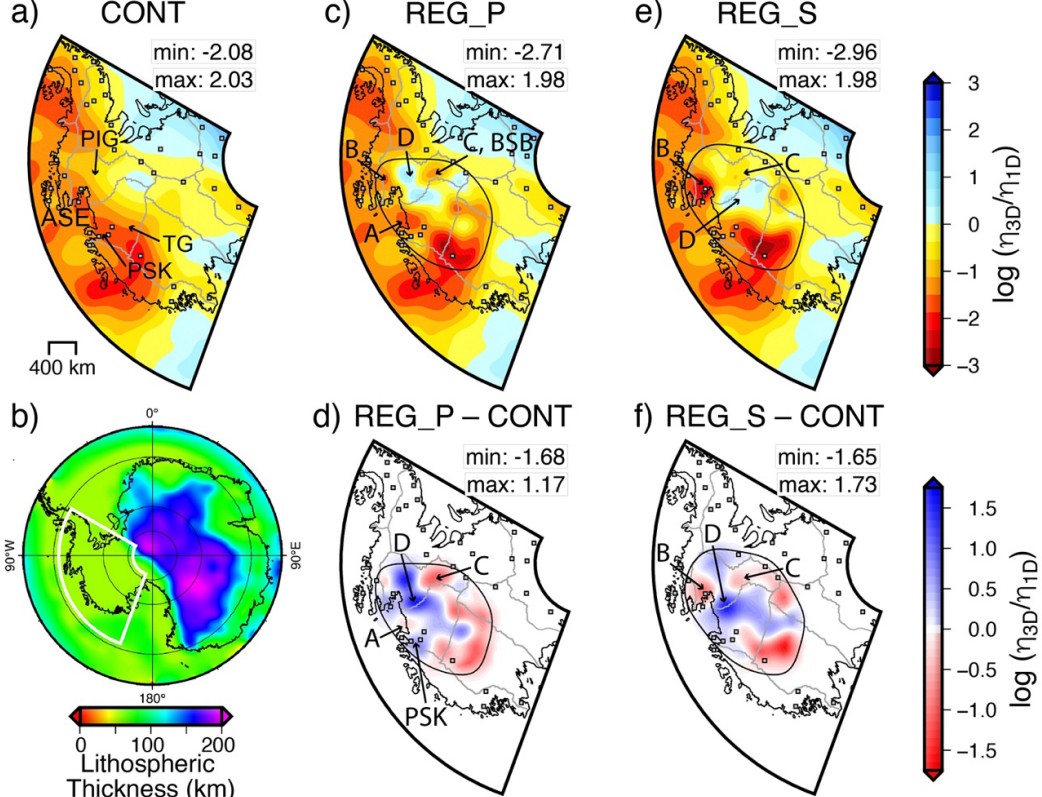

**Figure 1. Configuration of 3-D Earth models.** (a) Mantle viscosity variation at 150 km depth for the CONT viscosity model. Mantle viscosity variation is presented as the logarithm of mantle viscosity variations relative to the 1-D reference viscosity profile, which has viscosities of 5 x $10^{20}$ Pa s and 5 x $10^{21}$ Pa s in the upper and lower mantle, respectively. Thwaites Glacier and Pine Island Glacier are labeled TG and PIG, respectively. The region in which the Pope, Smith, and



Kohler glaciers are located is labeled PSK. Glaciers within the PSK are labeled individually in Figure 7c. The location of maps (a), (c-f) is outlined in white in (b). (b) Lithospheric thickness model of Wiens et al. (2021) for Antarctica. (c, e) Logarithmic viscosity perturbation maps at 150 km depth for the (c) REG_P and (e) REG_S viscosity models. The extent over which the regional seismic models are patched into the composite ANT-20 and GLAD-M25 seismic model is outlined

in solid black in (c-f). Panels (c) and (e) are annotated with the maximum and minimum logarithmic viscosity perturbation in the respective Earth model at 150 km depth. (d, f) Difference in logarithmic viscosity perturbations between the (d) REG_P and CONT viscosity models and the (f) REG_S and CONT viscosity model. Panels (d) and (f) are annotated with the maximum and minimum differences in logarithmic velocity perturbations between the (d) REG_P and (f) REG_S

and CONT viscosity models. The grounding line is delineated in black (Fretwell et al., 2013). POLENET-ANET and UKANET GPS station locations are plotted with squares in (a, c-f). Glacial drainage basin system boundaries are outlined in gray in (a, c-f) (Mouginot et al., 2017). The locations of upper mantle viscosity Features A, B, C, and D are labeled in (c-f). The location of upper mantle viscosity Feature C coincides with the location of the Byrd Subglacial Basin (BSB).


In constructing the 3-D Earth models, variations in mantle viscosity are estimated from relative variations in seismic velocity following Latychev et al. (2005a) and Austermann et al. (2013). Assuming that temperature is the only factor controlling seismic velocity variability, relative variations in seismic velocity are first converted to density, then temperature, and finally viscosity.

A scaling factor, $\epsilon$, is adopted in the conversion of temperature to viscosity variations. Lateral mantle viscosity variations are superimposed on a 1-D reference viscosity profile in the 3-D Earth models. The 1-D reference viscosity profile has viscosities of $5 \times 10^{20}$ Pa s and $5 \times 10^{21}$ Pa s in the upper and lower mantle, respectively, which is typical for most GIA-based inferences of mantle viscosity (e.g., Mitrovica & Forte, 2004) and consistent with previous GIA modeling efforts in

West Antarctica (e.g., Powell et al., 2020; Wan et al., 2022). Like previous 3-D GIA modeling studies, in Fig. 1 we plot 3-D viscosity variations relative to the 1-D reference viscosity profile. In Fig. 1, positive viscosity variations correspond to regions where upper mantle viscosity values in our 3-D viscosity models are greater than those found in the 1-D reference viscosity profile and the opposite holds for negative viscosity variations.






### 2.1.1 Continental-scale viscosity model (CONT)

The continental viscosity model, which will hereafter be referred to as the "CONT" model, is constructed by inserting the ANT-20 continental-scale shear-wave seismic model (Lloyd et al., 2020) into the GLAD-M25 global shear-wave seismic model (Lei et al., 2020) between the base

of the lithosphere and the transition zone in the region south of 47° S (Fig. 1a). With viscosity variations estimated from ANT-20 beneath Antarctica, the CONT model is similar to Gomez et al. (2024); however, global viscosity variations are inferred from the GLAD-M25 global seismic tomography model in CONT while Gomez et al. (2024) infer global viscosity variability from the S362ANI seismic tomography model (Kustowski et al., 2008). Global viscosity variations are

estimated from GLAD-M25 because it offers improved mantle structure resolution compared to S326ANI (Lei et al., 2020). Consistent with Gomez et al. (2024), a scaling factor of 0.033° $C^{-1}$ is adopted in Antarctica and 0.04° $C^{-1}$ for the global model in relating temperature to viscosity. The length-scale of features imaged in ANT-20 are limited by lateral and vertical smoothing factors employed in the tomographic inversion. Lateral features in the upper mantle and transition zone

are resolved at length-scales of ~140 km and ~340 km, respectively, and vertical smoothing is fixed at ~45 km for all depths (Lloyd et al., 2020).

### 2.1.2 Regional-scale viscosity models (REG_P, REG_S)

We construct two regional viscosity models (REG_P, REG_S) inferred from composite seismic

velocity models, in which the Lucas et al. (2020) P-wave and S-wave body wave seismic models of central West Antarctica (Fig. 1c-f) are inserted into the composite ANT-20 and GLAD-M25 seismic model described in Section 2.1.1. Several decisions must be made in constructing the REG_P and REG_S models, including the spatial and depth extent over which the Lucas et al. (2020) models are inserted into the composite model and how to correct Below, we describe the

approach adopted to construct the regional viscosity models.

The Lloyd et al. (2020) continental model and Lucas et al. (2020) regional models show comparable first-order upper mantle structure; however, with different tomographic approaches and datasets, discrepancies exist between the seismic tomography models (Figs. 1, S1). Different

from the adjoint tomography approach employed in Lloyd et al. (2020) to image upper mantle structure across Antarctica, Lucas et al. (2020) invert for upper mantle structure within just central



West Antarctica using a ray-path based travel-time tomography approach (VanDecar, 1991). The relative travel-time tomography approach employed to image regional upper mantle structure in Lucas et al. (2020) incorporates information from short-period body waves (1-25 second period), whereas the Lloyd et al. (2020) continental model only incorporates information from long-period body waves (30-80 second period) and surface waves (80-120 second period). As shorter-period seismic waves are more sensitive to finer-scale structures compared to longer-period waves, it is expected that shorter wavelength lateral variability in upper mantle structure is better resolved in the regional seismic tomography models compared to the continental seismic tomography model. The Lucas et al. (2020) study evaluates model resolution using a series of synthetic checkerboard tests, a standard procedure to test the resolution of body-wave tomography models. Between depths of 100-400 km, lateral structures ~100 km and ~200 km in length are resolved in the P-wave and S-wave models, respectively. As is common in body wave tomography, vertical resolution is limited in the Lucas et al. (2020) regional seismic models; however, resolution tests indicate that the imaged velocity anomalies primarily originate from mantle structure between the Moho and ~250 km depth.

We construct viscosity models using both the P- and S-wave regional seismic models of Lucas et al. (2020) because the models show somewhat different structure throughout central West Antarctica (Figs. 1, S1). The regional seismic tomography models are inserted into the ANT-20 model between the base of the lithosphere to 250 km depth in regions where the synthetic checkerboard tests are well-resolved. To evaluate the impact of inserting the regional seismic tomography models into the ANT-20 model over different depth extents, a series of simulations are performed that adopt viscosity models where the regional models are inserted into the continental base model between the base of the lithosphere and either 200 km or 300 km depth (Fig. S2). Up to ~5% difference is found between these simulations and those in which the regional seismic tomography models are inserted to 250 km depth, indicating that the depth extent over which the regional models are inserted into the ANT-20 model has a relatively minor impact on overall GIA model predictions (Fig. S2). Here and throughout this study, percent difference is calculated by taking the difference between predictions from two simulations of interest (i.e., prediction from simulation 1 – prediction from simulation 2) and dividing this difference by the prediction from simulation 2.





Due to regularization used in the tomographic inversion, seismic velocity anomaly magnitudes are
underestimated in the Lucas et al. (2020) models. Using the synthetic checkerboard resolution
tests, Lucas et al. (2020) estimates both low- and high-end amplitude recovery values for the P-
and S-wave models. Velocity amplitudes recovered in the P- and S-wave models range from ~20-
30% and ~15-25%, respectively. To further clarify, this means that a 1.0% P-wave velocity
anomaly in the seismic velocity model, for example, would correspond to a 3.3% - 5% P-wave
velocity anomaly in the mantle. Throughout this study we will discuss results from simulations
that adopt regional Earth viscosity models constructed assuming low-end amplitude recovery (i.e.,
~20% amplitude recovery for P-wave model and ~15% for S-wave model). The viscosity models
are constructed by first scaling the P-wave and S-wave models for low-end amplitude recovery
and then patching the scaled models into the composite ANT-20 and GLAD-M25 global seismic
model. We note that the P-wave model of Lucas et al. (2020) is converted to shear-wave velocity
anomalies prior to being inserted into the composite model. A simple conversion, where $\frac{\partial \ln (V_P)}{\partial \ln (V_S)} =$
0.4, is employed to convert P-wave velocity anomalies to S-wave velocity anomalies (Antolik et
al., 2003). The composite viscosity models incorporating regional upper mantle variations inferred
from the P-wave and S-wave models in central West Antarctica will hereafter be referred to as
"REG_P" and "REG_S", respectively. A scaling factor of 0.033° C$^{-1}$ is used in the conversion of
temperature to viscosity for both the REG_P and REG_S models. To assess the impact of
accounting for low-end versus high-end amplitude recovery in the body-wave models, we compare
results from simulations with Earth models constructed assuming high-end amplitude recovery
(i.e., 30% amplitude recovery for P-wave model and ~25% for S-wave model) with simulations
adopting REG_P and REG_S in Fig. S3. A similar spatial pattern of GIA predictions, albeit with
up to ~7% difference, is found between simulations adopting viscosity models constructed
assuming high-end versus low-end amplitude recovery (see Fig. S3 and caption for details).

### 2.1.3 Regional upper mantle viscosity model features

Here, we highlight key upper mantle viscosity features in the regional-scale viscosity models and
their relation to the geologic history and mantle dynamics in central West Antarctica (Features A,
B, C, D in Fig. 1c-f). Discrepancies in the spatial patterns and magnitudes of upper mantle viscosity
in the CONT, REG_P, and REG_S models can be attributed to differences in the seismic



tomography models used to construct each model. Feature A, most evident in REG_P but also

present to a lesser degree in the CONT and REG_S models, corresponds to a region with lower
viscosity upper mantle material located at the mouth of TG (Fig. 1c-d). Seismic imaging studies
have attributed this feature to warm mantle material flowing away from Marie Byrd Land (e.g.,
Lucas et al., 2020; Lloyd et al., 2020). Feature B, located beneath PIG, corresponds to a region
with lower upper mantle viscosity in REG_P and REG_S compared to CONT (Fig. 1c-f). Lower

viscosity upper mantle material beneath PIG is consistent with a range of processes that have
affected the region in the recent geologic past or continue to affect the region, including (1) warm
upper mantle material flowing from Marie Byrd Land, (2) focused Cenozoic extension (Jordan et
al., 2010), and/or (3) active subglacial volcanism (Corr & Vaughan, 2008; Quartini et al., 2021;
Geyer et al., 2021). Low viscosity Feature C, most prominent in REG_P, is located beneath a

portion of the Byrd Subglacial Basin (Fig. 1c-f), a deep ice-filled graben in a region that likely
underwent localized Cenozoic extension (e.g., Jordan et al., 2010; Chaput et al., 2014; Lloyd et
al., 2015; Lucas et al., 2020). Finally, high viscosity Feature D, located in the interior of the
Thwaites and Pine Island glacier drainage basins, is evident in both the REG_P and REG_S
viscosity models (Fig. 1c-f). Higher viscosities in the interior of central West Antarctica are

consistent with relatively cool upper mantle material that has remained unperturbed by tectonic
activity since the major extensional phase of the West Antarctic Rift System in the late Cretaceous
(e.g., Siddoway, 2008; Lucas et al., 2020; Lloyd et al., 2020).

### 2.1.4  On viscosity model uncertainty

The resolution and accuracy of the 3-D viscosity models in our study is intrinsically dependent
upon the seismic tomography models. While the Lucas et al. (2020) body wave tomography
models capture shorter wavelength lateral variability in upper mantle structure compared to the
Lloyd et al. (2020) model, there is still significant uncertainty associated with the pattern and
magnitude of lateral variations in upper mantle seismic velocities throughout West Antarctica.

There is also uncertainty associated with the adopted radial viscosity variability and the
quantitative relations and assumptions used to convert seismic tomography models to viscosity
(e.g., Ivins et al., 2023). With recent work showing the impact of transient rheology on GIA
predictions (e.g., Adhikari et al., 2021; Ivins et al., 2022; Lau, 2023), we also acknowledge
potential error associated with the adoption of Maxwell rheology in our modeling approach.





Overall, given the substantial Earth model uncertainty, we emphasize that the intent of this study is to evaluate the degree to which laterally varying upper mantle viscosities have the potential to impact GIA predictions, and the viscosity models adopted here adequately serve this goal. We do not argue for the accuracy of GIA predictions made using one viscosity model over another; instead, we intend to compare predictions made using various viscosity models that capture

different degrees of heterogeneity in upper mantle structure.

### 2.2 Ice Models

We adopt ice models to represent modern (1992 – 2017), extended modern (1892 – 2017), and future (2000-2300) ice loading scenarios in Antarctica (Fig. 2). The modern ice model, which will

be referred to as ICE-25 throughout this study, is constructed based on the Shepherd et al. (2019) time series of surface elevation change ($\Delta h$) across the Antarctic Ice Sheet between 1992 and 2017 (Fig. 2a). The Shepherd et al. (2019) $\Delta h$ time series is derived from multi-mission satellite altimetry data and is resolved over a 5 km grid at 5-year time intervals. Following the same procedure as Wan et al. (2022), we construct the modern ice model used in the GIA simulations

by treating $\Delta h$ as a proxy for ice thickness change (Carrivick et al., 2019). The initial ice thickness and bedrock topography across Antarctica is based on Bedmap2 (Fretwell et al., 2013), noting that results are relatively insensitive to this choice since minimal grounding line changes occur during this time frame. A grounded ice mask is constructed using the Bedmap2 grounding line extent, and ice thickness changes >20 m/year are saturated to mitigate for spurious data.


In addition to the 25-year modern ice model, we construct an extended modern ice history (ICE-125) for 125 years, from 1892 – 2017 (Fig. 2b). This extension is motivated by evidence for upper mantle viscosities between ~$10^{18}$ to $10^{19}$ Pa s in the ASE and the GIA response time scales ranging from decades up to a century (e.g., Barletta et al., 2018). From 1992 – 2017 in ICE-125, we adopt

the same ice history as that in the ICE-25 model. Limited observations exist to constrain ice history prior to the satellite altimetry era; therefore, we construct an ice model using the pattern and rate of ice change between 1992 – 2002 from the ICE-25 model rescaled by a factor of 25% for the period between 1892 and 1992 (Fig. 2b). This is similar to the approach used by Barletta et al. (2018) to build an ice history for 1900 – 2014. Like Barletta et al. (2018), we adopt a 25% scaling

factor as it is compatible with studies that show a slower ice flow rate between 1970 – 2002 relative



to the present day (Mouginot et al., 2014). An assumption is made that the lower flow rate is consistent with a lower ice mass loss rate from 1892 – 1992.

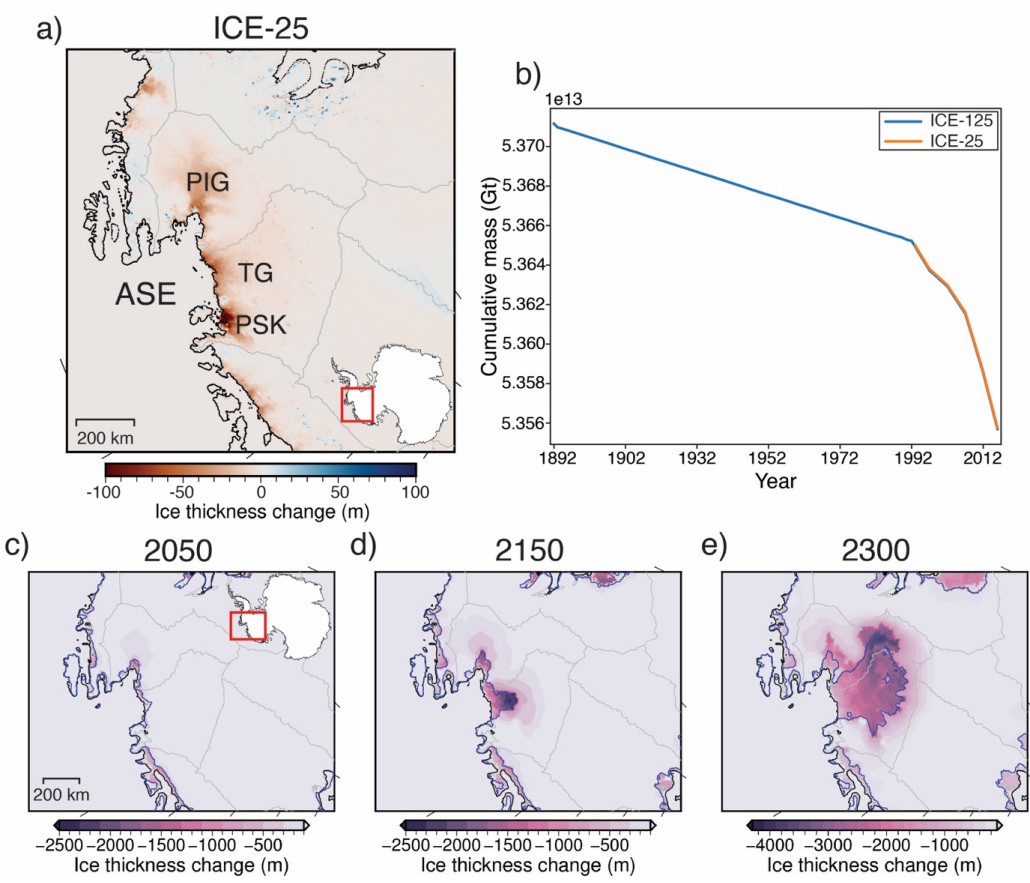

**Figure 2. Ice models:** (a) Total ice thickness change in meters from 1992 to 2017 for the observation-based ICE-25 ice model (Shepherd et al., 2019). (b) Cumulative mass in Gt for ice history ICE-25 between 1992 and 2017 and ICE-125 between 1892 and 2017 in central West Antarctica. As described in the text, the ICE-125 model is constructed using the ICE-25 model for 1992 – 2017. In ICE-125 for 1892 – 1992, the rate of ice change is assumed to be 25% of that in the ICE-25 model between 1992 – 2002. Figure 7c includes a map of total ice thickness change for the ICE-125 model like that plotted for ICE-25 in (a). (c-d) Total ice thickness change in (c) 2050, (d) 2150, and (e) 2300 for the ICE-FUT ice model.





In addition to the modern ice loading model, we produce GIA simulations with an Antarctica-wide
ice sheet model projection from Gomez et al. (2024) to represent future ice thickness changes (Fig.
2c-e). Using an ice sheet-sea-level coupling approach, Gomez et al. (2024) model AIS thickness
change and global GIA from 1950 to 2500 under a range of climate forcings and ice physics
assumptions. Here, we utilize an ice-sheet projection produced from a simulation performed with
the Representative Concentration Pathway (RCP) 2.6 climate forcing that incorporates the effects
of hydrofracturing and mechanical failure of marine-terminating ice cliffs (DeConto et al., 2021).
The Gomez et al. (2024) model predicts a contribution of the West Antarctic Ice Sheet to global
mean sea level rise of 0.33 m by 2150 and 1.23 m by 2300, with significant ice loss and grounding
line retreat in the ASE. This ice model projection will be referred to as ICE-FUT hereafter. The
ICE-FUT ice model simulation has a continent-wide resolution of 10 km, with 5 km resolution in
West Antarctica, and ice thickness is provided to the GIA model at 2-year time intervals from
1950-2500. We adopt the same initial bedrock topography, Bedmap2 (Fretwell et al., 2013), as
that used in Gomez et al. (2024).

### 2.3   Crustal motion rates observed at POLENET-ANET GPS sites

In section 4.1, we compare vertical and horizontal crustal motion rates predicted in our GIA
simulations to crustal motion rates observed at select POLENET-ANET GPS sites located
throughout central West Antarctica. The Antarctic GPS data were processed within a global
network composed of ~2500 stations (with data spanning from 1993 to 2022, ~4 million station-
days) using a parallelized Python wrapper for GAMIT/GLOBK v10.71 (Gómez, 2017). Processing
of GPS data used the orbits and antenna calibration parameters available from the International
GNSS Service (IGS14 reference frame), the Vienna Mapping Functions (Boehm et al., 2006) to
estimate the atmospheric delays, and the ocean tide loading model FES2014b (Lyard et al., 2021).
We use an automated procedure to fit trajectory models to the displacement time series of each
CGPS station (Bevis & Brown, 2014; Bevis et al., 2019). Reference frame (RF) realization and
trajectory modeling are implemented simultaneously, to ensure internal geometrical consistency
(Bevis & Brown, 2014). The horizontal aspect of the RF, in velocity or rate space, is imposed by
minimizing the RMS horizontal velocities of a set of stations considered to be part of the rigid
portions of the Antarctica plate, in which there are no relative velocities driven by tectonics. The
vertical aspect of the RF in velocity space is that which minimizes the RMS vertical velocities of





370 a global set of CGPS stations called VREF, chosen using the "ensemble of RFs" approach described by Bevis et al. (2013). There are 15 HREF stations on the Antarctic continent and 1 on Kerguelen Island. The RMS horizontal velocity of these stations in the final ANET frame is 0.29 mm/yr. The RMS vertical velocity of the 850 VREF stations is 0.92 mm/yr. The station displacement time series and best-fit trajectory models referred to this RF are denoted as the

375 geodetic solution pg03f_PC_H16.

## 3 Results

The goal of this analysis is to assess how regional-scale variations in upper mantle viscosity impact GIA predictions in central West Antarctica in response to modern and future ice loading. We start

380 by briefly summarizing key upper mantle features in the viscosity models. Then we compare the predicted crustal deformation rates, sea level changes, and geoid changes from simulations adopting the CONT, REG_P, REG_S, and 1D viscosity models for ice mass changes over the past 125 years (ICE-125). Following this, we assess the impact of adopting a longer 125-year ice history (ICE-125) versus a shorter 25-year ice history (ICE-25) on GIA predictions. Finally, we evaluate

385 predictions of sea level and bedrock elevation change from simulations adopting the CONT, REG_P, and REG_S, and 1D viscosity models with the ICE-FUT ice sheet projection.

### 3.1 Results with the continental viscosity model (CONT) and modern ice mass changes (ICE-125)

390 For all simulations, regardless of the adopted viscosity model, relative sea level fall is predicted over most of the study region because of viscoelastic bedrock uplift and geoid subsidence in response to ice mass loss, with the former being the dominant signal (Fig. 3). Earth rotational effects are negligible compared to other effects in the vicinity of ice mass loss. For simulations with the CONT viscosity model, up to 87 cm of sea level fall is predicted in the ASE at the end of

395 the 125-year simulation (Fig. 3a). Peak sea level fall and crustal uplift rates are found in the region of the Pope, Smith, and Kohler glaciers (PSK), which is situated in the Thwaites Glacier drainage basin as shown in Fig. 2, coincident with the region of greatest ice mass loss in the ICE-125 ice history (Figs. 2a, 3a, d). Horizontal crustal motions generally point outward from the region of greatest ice mass loss in the TG, PIG, and PSK regions (Fig. 3g). Peak predicted horizontal crustal

400 rates (11.45 mm/year) are localized in the eastern Thwaites Glacier basin, ~60 km inland from the





grounding line (Fig. 3g). In agreement with previous work (Powell et al., 2020; Wan et al., 2022), the addition of continental-scale 3-D variations in viscosity structure results in GIA predictions that diverge from simulations adopting the 1D model (Fig. S4). The largest discrepancies in GIA predictions between the simulations adopting the 1D and CONT viscosity models are found in the

TG basin, where higher relative sea level and lower magnitude vertical crustal rates are found for simulations adopting the 1D viscosity model (Figs. S4, S5).

## 3.2 Results with the regional viscosity models (REG_P, REG_S) and modern ice mass changes (ICE-125)

Comparing continental simulations to those adopting the REG_P and REG_S viscosity models, it is evident that regional-scale upper mantle viscosity variability impacts GIA predictions associated with modern ice mass changes (Figs. 3, S5). The largest discrepancies between sea level and crustal motion predictions amongst simulations adopting CONT, REG_P, and REG_S are found proximal to the grounding line in central West Antarctica (Figs. 3, S5).


In the PSK region (labeled in Fig. 1d), ~1 - 5.4 cm less sea level fall is predicted for simulations adopting REG_P and REG_S compared to the simulation with the CONT viscosity model (Fig. 3b-c). Compared to the CONT model, the PSK region is largely underlain by higher viscosity upper mantle in both REG_P and REG_S (Fig. 1c-f), contributing to higher relative sea level

predictions. ~5-20% slower vertical crustal uplift rates are also predicted with the regional viscosity models in this region at the end of the 125-year modern ice loading history (Fig. 3e-f). Within central West Antarctica, the greatest discrepancies in predictions of horizontal crustal motions between simulations adopting the regional versus continental viscosity models are focused in the PSK region (Fig. 3h-i). Up to ~5 mm/year difference in horizontal crustal rates are found

between simulations adopting the REG_P and REG_S models compared to the CONT model (Fig. 3h-i).

Now to focus on the TG and PIG regions, in both the CONT and REG_P viscosity models, there is a particularly low viscosity upper mantle feature that extends from the coastal PSK region across

the grounding line of the TG (Fig. 1a, c). While this viscosity feature extends further east along coastal ASE beneath the PIG with a relatively constant magnitude in CONT, stronger spatial



variations in viscosity are found across the TG and PIG in REG_P (Fig. 1a, c, d). Notably, compared to CONT, the central portion of the TG grounding line is underlain by lower viscosity upper mantle material in REG_P (Feature A; Fig. 1c-d). The inclusion of Feature A may contribute
to slightly more sea level fall evident adjacent to the central portion of the TG grounding line in simulations with REG_P compared to CONT (Fig. 3b). On the other hand, higher viscosities more broadly across the TG basin in the REG_P model compared to CONT (Fig. 1d) produce slower present-day uplift rates across the TG and PIG grounding lines (Fig. 3e). While faster uplift rates may be expected in a region with higher sea level fall, relative sea level predictions reflect variation
over the entire simulation period whereas present-day uplift rates represent a snapshot in time at the end of the simulation period. Discrepancies in horizontal motion predictions between CONT and REG_P models occur across notably shorter spatial length-scales across the TG and PIG grounding line region compared to predictions of vertical motion and sea level change (Fig. 3h).

Unlike the REG_P viscosity model, the REG_S model includes higher viscosity upper mantle material across the mouth of TG which results in up to 5.4 cm (or ~10-15%) less sea fall in the region compared to simulations with CONT (Fig. 3c). As the resolution of the P-wave tomography model used to construct the REG_P viscosity model is notably better than the resolution of the S-wave model in the TG grounding line region (Lucas et al., 2020), predictions made using the
REG_P model are likely to be more reliable than predictions made using the REG_S model in this region. Compared to simulations with CONT, a low viscosity upper mantle feature prominent in REG_S beneath PIG (Feature B; Figs 1e-f) produces up to 6.5 cm more sea level fall across the modern simulation throughout the PIG region (Fig. 3c) and up to 2.9 mm/year faster crustal uplift rates at the end of the simulation (Fig. 3f). Across much of the PSK, TG, and PIG grounding line
region, a ~0.5 – 4.0 mm/year difference in horizontal crustal motion predictions is found between simulations adopting REG_S versus CONT (Fig. 3i). Similar to the REG_P simulation, we find strong discrepancies in predictions of horizontal crustal motions over short length-scales (~50-100 km) between simulations adopting the REG_S and CONT viscosity models.





**Figure 3. Influence of regional upper mantle structure on predictions of relative sea level and crustal motion rates for modern ice loading (ICE-125).** (a) Total sea level change for a GIA simulation with the CONT viscosity model and the ICE-125 ice model. (b-c) Difference in predicted relative sea level between simulations adopting the (b) REG_P and CONT viscosity





models and the (c) REG_S and CONT viscosity models. (d) Predicted vertical crustal rates at the end of the 125-year simulation with the CONT viscosity model. (e-f) Difference in predicted vertical crustal rates between simulations adopting the (e) REG_P and CONT viscosity models and the (f) REG_S and CONT viscosity models. (g) Horizontal crustal rates predicted at the end of the simulation adopting the CONT viscosity model. (h-i) Difference in horizontal crustal rates

after 125 years of loading between the (h) REG_P, (i) REG_S and CONT viscosity models. Color contours represent the difference in the predicted magnitude of horizontal crustal rates between simulations adopting the (h) REG_P and CONT viscosity models and (i) REG_S and CONT viscosity models. Vectors show the difference in the predicted direction and magnitude of horizontal crustal rates between the respective panel's viscosity model and the CONT viscosity

model. Black and purple arrows correspond to locations where horizontal crustal rate differences are ≥1 mm/year and <1 mm/year, respectively. Extent of maps area shown in (a). The locations of upper mantle viscosity Features A, B, C, and D are labeled in (b).

### 3.3  Comparison of results with 125-year versus 25-year ice histories

Several studies investigating GIA in central West Antarctica have adopted 25-year ice histories to model the viscoelastic response of the solid Earth to contemporary ice mass changes (e.g., Powell et al., 2020; Wan et al., 2022; Powell et al., 2022) With upper mantle viscosities suggested in the literature ranging from ~4 x $10^{18}$ Pa s up to $10^{20}$ Pa s in central West Antarctica (e.g., Nield et al., 2014; Barletta et al., 2018; Ivins et al., 2023), viscous effects are expected to be significant on

decadal to centennial timescale (Hay et al., 2017; Kachuck et al., 2020; Wan et al., 2022). Therefore, a 25-year ice history may not be long enough to capture the entire viscous response to contemporary ice mass change. Here, we assess the impact of extending the length of our modern ice history from 25 to 125 years on GIA predictions. Figure 4 shows the differences in predicted relative sea level and geoid changes over the last 25 years and current crustal motions between

simulations adopting the ICE-125 and ICE-25 ice models for the REG_P viscosity model. Because the ICE-125 and ICE-25 models adopt the same ice loading history for 1992–2017, Figure 4 represents the impact of viscous deformation due to ice loading changes between 1892 – 1992 on predicted relative sea level, crustal motion and geoid height during the 25-year record time frame. Analogous comparisons with the CONT, REG_S, and 1D viscosity models predicting similar

magnitude differences are shown in Fig. S7.





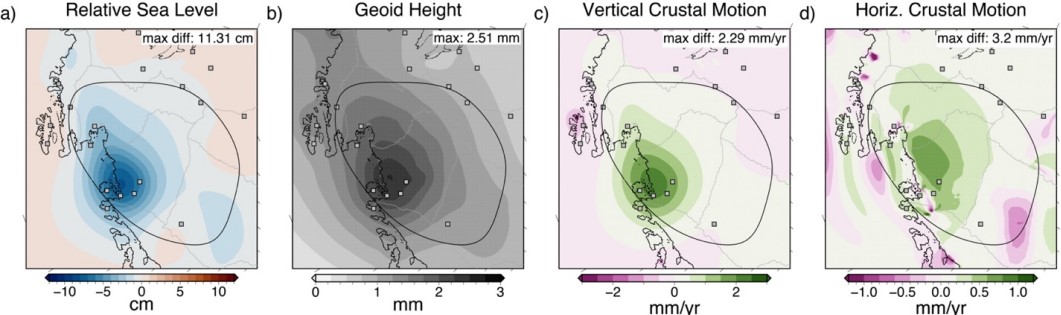

**Figure 4. Impact of modern ice history length (i.e., 125-years versus 25-years) on model predictions.** (a-b) Difference in (a) relative sea level and (b) geoid height changes from 1992 –
2017 between simulations adopting the ICE-125 and ICE-25 ice histories with the REG_P viscosity model. (c-d) Difference in (c) vertical crustal rates and (d) horizontal crustal rates at the end of the simulations in 2017 with the REG_P viscosity model. Extent of maps shown in all panels are the same as the inset in Figure 2a.

Overall, the greatest discrepancies between relative sea level and geoid height change predictions between simulations adopting ICE-125 and ICE-25 are concentrated near the grounding line across the PSK and TG regions (Fig. 4), where the greatest ice loading changes occur prior to 1992 (Fig. 2a). Indicative of an ongoing viscous contribution from ice mass changes between 1892-1992, up to 11.3 cm greater sea level fall (~50% of total signal with ICE-125) and 2.5 mm difference in
predicted geoid height change (~8% of the total signal with ICE-125) are found in simulations adopting ICE-125 versus ICE-25 model between 1992 and 2017 (Fig. 4a-b). Regardless of the adopted viscosity model, vertical crustal motion rates computed at the end of the ICE-125 ice history are greater in magnitude (up to ~2.8 mm/year) compared to those computed with the ICE-25 ice history (Figs. 4c, S7). Discrepancies between horizontal crustal motions rates in simulations
adopting ICE-125 versus ICE-25 are more spatially variable, with notable differences found adjacent to the grounding line (Fig. 4d). While the viscous contribution to GIA predictions from ice mass changes between 1892 – 1992 is notable throughout much of the TG and PIG basins, the contribution becomes negligible ~500 km inland of the modern-day grounding line.




### 3.4 Incorporating regional upper mantle viscosity into future GIA simulations

Figures 5 and 6 compare predictions of sea level change and bedrock elevation profiles with continental and regional viscosity models due to projected ice cover changes from 1950 to 2300 adopting the ICE-FUT ice sheet model projection (Gomez et al., 2024). As expected, regions
undergoing projected ice mass loss experience uplift and sea level fall in the vicinity of the grounding line in all simulations (Fig. 5). The differences between results with the regional and continental viscosity models can be understood by considering the combined spatial patterns of ice mass loss (Fig. 2c-e) and mantle viscosity differences (Fig. 1d, f).

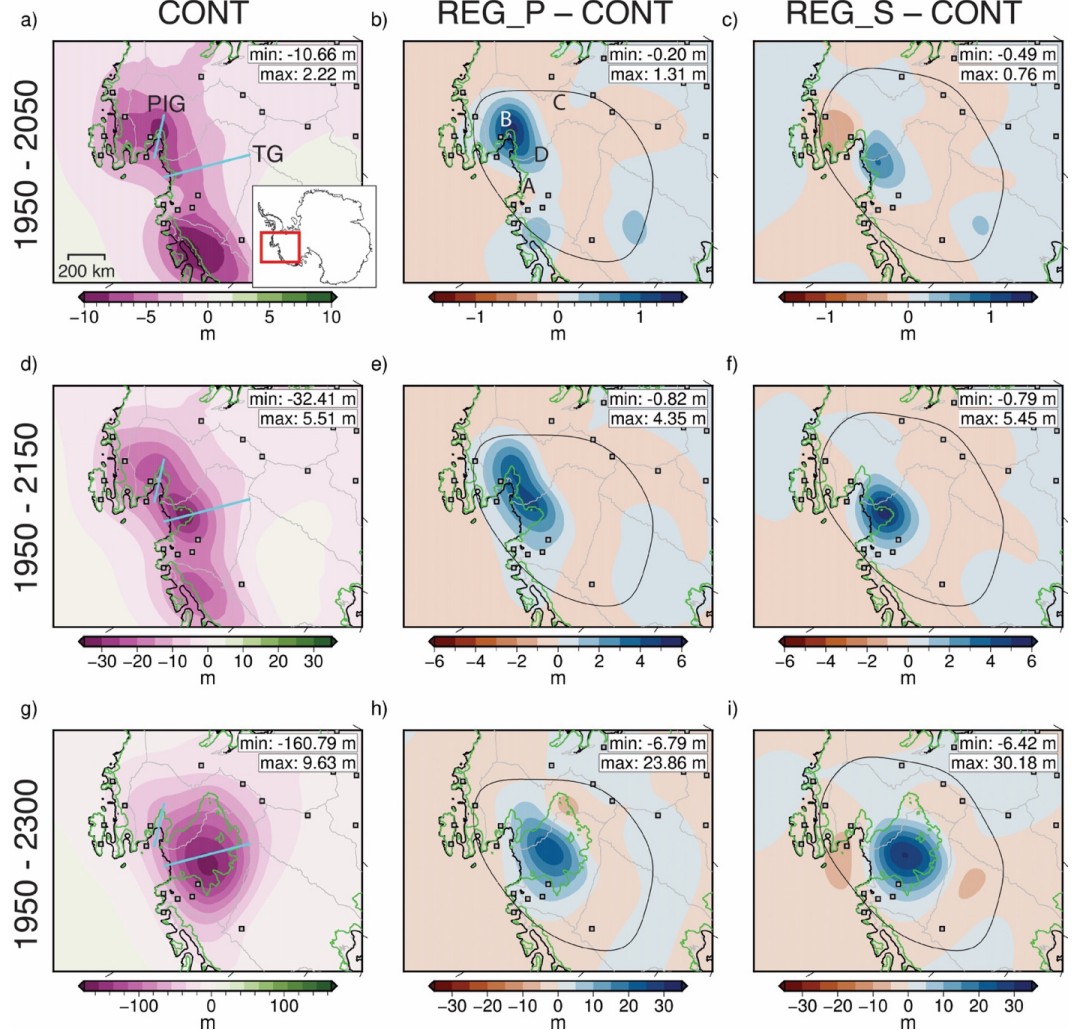





**Figure 5. Impact of regional upper mantle structure on relative sea level predictions for the ICE-FUT ice sheet projection.** (a, d, g) Total predicted sea level change at (a) 2050, (d) 2150, and (g) 2300 for GIA simulations with the CONT viscosity model. The current and predicted grounding line positions are shown in black and green, respectively (Gomez et al., 2024). (b, e, h)
Difference in predicted sea level change between simulations adopting the REG_P and CONT viscosity models for (b) 2050, (e) 2150, and (h) 2300. (c, f, i) Difference in predicted sea level change between simulations with the REG_S and CONT viscosity models for (c) 2050, (f) 2150, and (i) 2300. The location of profiles across the TG and PIG shown in Figure 6 are plotted the first column. Extent of maps in all panels is shown in (a). The locations of upper mantle viscosity
Features A, B, C, and D are labeled in (b).

In 2050, simulations with REG_P and REG_S produce distinct patterns of relative sea level change throughout the ASE (Fig. 5a-c). Compared to simulation with CONT, higher relative sea level (+1.31 m compared to CONT) is predicted in the central PIG basin with the REG_P model while
lower relative sea level (-0.49 m compared to CONT) is predicted in the northern PIG basin with the REG_S model (Fig. 5a-c). These discrepancies in relative sea level predictions in the PIG basin can be attributed to low viscosity Feature B, which is most prominent in REG_S (Fig. 1c, e), contributing to more crustal uplift in the region. While only slight variability is found in relative sea level predictions throughout the TG basin between simulations adopting REG_P versus CONT
in 2050 (Fig. 5b), up to ~0.8 m higher relative sea level is predicted at the TG grounding line in the simulation with REG_S compared to CONT (Figs. 5c, 6c).

A transition from lower upper mantle viscosities to higher viscosities moving from the grounding line towards the interior of the TG and PIG basins is found in the CONT model (Fig. 1a). Compared
to CONT, this viscosity transition is much sharper, occurring over a significantly shorter distance in the REG_P and REG_S models (Fig. 1c-e, d-f). Including the ~1 order of magnitude upper mantle viscosity transition over ~100 km in simulations with the REG_P and REG_S models has a notable impact on the spatial pattern and magnitude of sea level change predictions in the TG and PIG basins, resulting in ~10-17% higher sea level predictions at and adjacent to the grounding
line as soon as 2150 (compare REG_P, REG_S profiles to CONT profile in Fig. 6e-f; Fig. 5e-f).



In 2300, the largest difference in predicted sea level change between simulations adopting the regional (REG_P, REG_S) versus continental viscosity models is focused in a region interior to the modern TG catchment over which the TG grounding line has migrated during the simulation (Fig. 5h). Up to ~25% less sea level fall is predicted by 2300 in simulations with the regional viscosity models compared to the continental model (Fig. 5h). The location in which higher sea level is predicted in simulations adopting the regional viscosity models (Fig. 5h-i) coincides with the location of a high viscosity upper mantle feature in both the REG_P and REG_S viscosity models (Feature D, Figs. 1, 5b). The effect of including high viscosity Feature D is evident across the TG profile in Fig. 6g, which shows lower bedrock elevation change predicted in simulations with REG_P and REG_s versus CONT. Additionally, best resolved in REG_P, a localized low viscosity upper mantle feature, coincident with the location of the Byrd Subglacial Basin (Feature C; Figs. 1c, 5b), also appears to produce up to ~10 m of additional sea level fall at the eastern portion of the TG grounding line by 2300 (Fig. 5h).

## 4 Discussion

Our simulations indicate that regional-scale lateral variability in upper mantle viscosity has spatially variable impacts on the rate and magnitude of the solid Earth response to modern and future ice mass changes in central West Antarctica. We adopt central West Antarctica as our study location because it is a region characterized by strong lateral variability in upper mantle structure and is undergoing active marine ice sheet retreat that is projected to continue. However, our findings may provide useful guidance to future studies that aim to evaluate solid Earth deformation in response to ice mass changes in other regions underlain by strongly varying Earth structure, such as Greenland, Alaska, Patagonia, and other sectors of Antarctica. In this section, we will start by discussing the implications of our findings for the interpretation of crustal motion rates observed at select GPS sites deployed across central West Antarctica and then proceed to discuss implications of incorporating regional-scale upper mantle structure into future GIA projections.



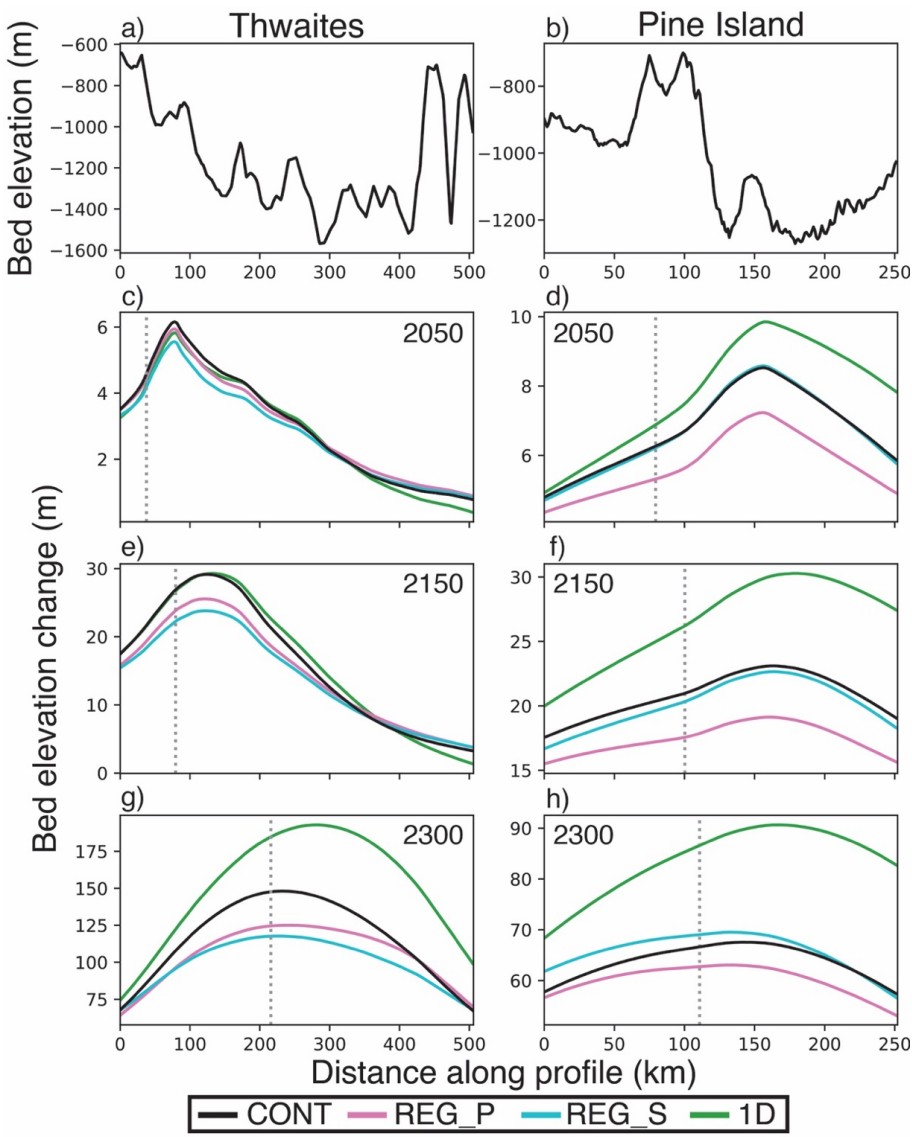


**Figure 6. Bedrock elevation change predicted at profiles across the Thwaites Glacier and Pine Island Glacier with the ICE-FUT ice sheet model projection.** (a) Bedrock elevation profile moving upstream TG shown in Fig. 5a. (b) Bedrock elevation profile moving upstream PIG plotted in Fig. 5a. (c-h) Total bedrock elevation change in meters predicted along the (c, e, g) TG profile

and (d, f, h) PIG profile for the CONT, REG_P, REG_S, and 1D viscosity models for 2050, 2150, and 2300. The location of the projected grounding line for 2050, 2150, and 2300 from the ICE-FUT ice sheet model projection is marked with a dashed gray line in panels (c-h).



### 4.1 Implications for the interpretation of bedrock motion observations

In Figure 7, we compare our predicted crustal motion rates to observed rates at GPS sites in the study region (see Methods section 2.3, black dots in Figure 7). Note that the solid Earth has a multi-normal mode response to surface loading (Peltier, 1974), and while very low viscosity zones in the uppermost mantle may have largely finished responding to Late Pleistocene and Holocene ice loading signals by the modern era, ongoing viscous deformation at other modes remains and

must be considered (i.e. what is often referred to as a "GIA correction" in the literature must be made to the observed rates before comparing them to our model predictions for the modern). Estimates of vertical crustal motion resulting from ice mass changes since the Last Glacial Maximum (LGM) differ widely in the literature (e.g., see Fig. 2 in Whitehouse et al., 2019), but within central West Antarctica, most studies predict <5 mm/year contribution to vertical crustal

rates (e.g., Argus et al., 2014; van der Wal et al., 2015; Whitehouse et al., 2012; Ivins et al., 2013; Peltier et al., 2015; Gomez et al., 2018). Along with uncertainty associated with ice loading changes throughout the Late Pleistocene and Holocene, significant uncertainty is also associated with the more recent evolution of the West Antarctic Ice Sheet over the millennia preceding the satellite altimetry era. Some work suggests that ice sheet retreat and subsequent readvance affected

the ASE in the Holocene (Balco et al., 2023), while others argue against retreat and readvance in the Holocene (Clark et al., 2024). Here we note that ongoing work, which applies Fréchet derivatives to the GIA problem (Lloyd et al., 2024), is focused on better characterizing the sensitivity of crustal rate observations in West Antarctica to past ice loading (Powell et al., 2023). Notwithstanding the substantial uncertainty, here we remove the contribution from ice changes

prior to the modern era from the observed vertical crustal rates using two differing estimates from the literature: the Gomez et al. (2018) model and the ICE-6G_C model of Peltier et al. (2015) (light and dark grey dots in Figure 7a). These models are chosen because they represent low- and high-end predictions of vertical crustal rates compared to other GIA models (e.g., Whitehouse et al., 2019). Note that neither model attempts to accurately treat late-Holocene ice mass changes.




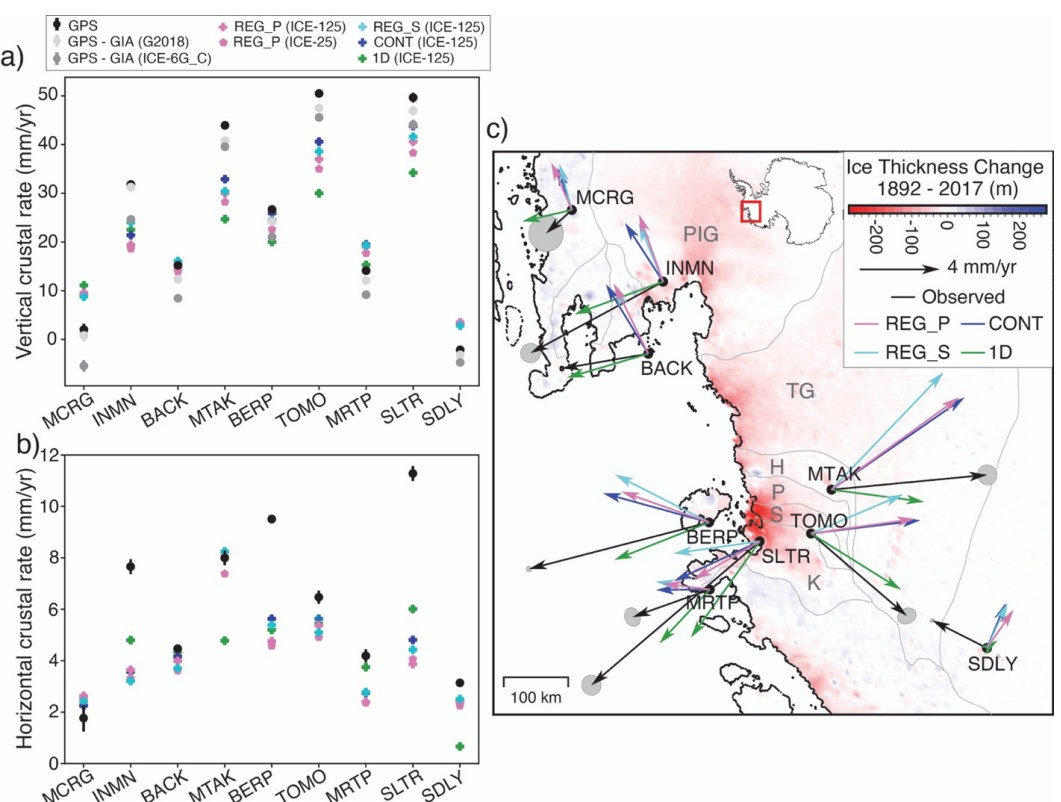

**Figure 7. Comparison between predicted and observed crustal rates for modern ice loading.**
(a) Vertical crustal rates observed at selected GPS sites located throughout central West Antarctica
plotted with predicted vertical crustal rates from simulations adopting the ICE-125 ice model with

the REG_P, REG_S, CONT, and 1D viscosity models. Predicted crustal rates are calculated as the
average crustal rate over the last five years of each simulation. Observed vertical crustal rates are
corrected based on the predictions from the Gomez et al. (2018) (abbreviated G2018 in the legend)
and ICE-6G_C (Peltier et al., 2015) models. Along with predictions for simulations with ICE-125,
crustal rate predictions for a simulation with ICE-25 and the REG_P viscosity model are also

shown. (b) Horizontal crustal rates observed at GPS sites plotted with predicted horizontal crustal
rate predictions. Refer to the legend in (a) for details on which symbols correspond to which model
runs. (c) Observed and predicted horizontal crustal rates plotted atop the ICE-125 extended modern
ice loading history. Refined glacier basin outlines of the Haynes, Pope, Smith, and Kohler Glaciers,
respectively labeled H, P, S, and K, are shown in (c) (Mouginot et al., 2017).




Discrepancies exist between the vertical crustal rates observed at GPS sites across central West Antarctica and those predicted in our GIA simulations (Fig. 7a; Table S1). Regardless of the adopted viscosity model, observed vertical crustal rates are under-predicted by GIA simulations at INMN, MTAK, TOMO, and SLTR, however, over-predicted at MCRG, BACK, MRTP, and
SDLY (Fig. 7a). Compared to simulations with 1D viscosity models, accounting for heterogeneous upper mantle structure, whether it be at the continental- or regional-scale, generally leads to the greatest reduction in discrepancies between predicted and observed vertical crustal rates (Fig. 7a; Table S1). Up to ~3.6 mm/year difference in vertical crustal rates is found amongst simulations adopting regional- versus continental-scale viscosity structure at GPS sites. We find that
introducing regional variations in viscosity reduces differences between predicted and observed vertical crustal rates at a number of GPS sites (INMN, BERP, MRTP, SDLY). For example, low viscosity Feature B prominent in the REG_S model in the PIG basin (Fig. 1e-f) likely produced a faster uplift rate at INMN (+24.0 mm/year), which is closer to the observed rate, compared to simulations adopting the REG_P and CONT viscosity models (+21.4 mm/year and +19.4 mm/year
at INMN, respectively) (Fig. 7a).

Discrepancies between observed and predicted horizontal crustal rates vary significantly with the adopted viscosity model and from site to site (Fig. 7b-c). Note that horizontal crustal rate observations are not adjusted for horizontal motion associated with ice mass changes following
the LGM given the great uncertainty associated with such predictions. Unlike with the vertical rates, adopting realistic viscosity structure does not always reduce discrepancies between the magnitude and direction of predicted and observed horizontal crustal rates (Fig. 7b-c). In fact, the direction of horizontal crustal rates predicted using the 1D viscosity model generally align better with the observed horizontal crustal rates than those predicted in simulations with 3-D viscosity
models (Fig. 7c).

While previous work has successfully modeled the direction and magnitude of observed horizontal crustal rates using 1-D GIA models for the ASE (Barletta et al., 2018), horizontal crustal rates predicted by 3-D GIA models have never been compared to observed rates in the region.
Comparing predictions from GIA simulations adopting 1-D and 3-D Earth models for modern ice mass changes (1992 – 2017) in West Antarctica, Powell et al. (2020) show that 3-D Earth structure



is necessary to model horizontal crustal rates and capture the viscous signal driving horizontal deformation. However, Powell et al. (2020) did not compare predicted and observed horizontal crustal rates in their analysis. On the other hand, from analyses with a 1-D GIA model and a

modern ice history for 1900 – 2014, Barletta et al. (2018) model observed horizontal crustal rates in the ASE relatively well and finds that horizontal crustal motion predictions are strongly dependent on the geometry of the ice load, more so than vertical crustal motions. From previous work in Antarctica and elsewhere (e.g., Kaufmann et al., 2005; Latychev et al., 2005b; Steffen et al., 2006; Hermans et al., 2018; Vardić et al., 2022), it is evident that horizontal crustal motions

depend both on the spatial pattern of ice mass changes and solid Earth structure. Like this study, other studies have also found greater discrepancies between observed and predicted horizontal crustal rates with the adoption of more complex, laterally varying Earth structure (e.g., Steffen et al., 2006). It is possible that inaccuracies in the 3-D viscosity models and ice history adopted in this study compound, in a sense, to produce the discrepant predictions of observed horizontal

crustal rates.

Overall, we find that accounting for regional-scale viscosity structure is likely necessary to accurately interpret vertical crustal rate observations at GPS sites, particularly in regions with localized low/high viscosity upper mantle features that are not well-resolved in continental-scale

viscosity models. On the other hand, interpretations for horizontal crustal motions are substantially more complex, and our results suggest that introducing 3-D structure may not improve these predictions without more accurate ice history models. These findings have implications for studies that estimate lithospheric thickness and upper mantle viscosity using bedrock motion observations (e.g., Nield et al., 2014; Wolstencroft et al., 2015; Zhao et al., 2017; Barletta et al., 2018; Samrat

et al., 2021). Because regional, basin-scale (~50-100 km) variability in upper mantle viscosity will impact GIA predictions, caution is warranted in estimating upper mantle viscosity using 1-D GIA models in concert with observations of bedrock motion. Finally, improving constraints on ice history is critical for reducing data-model misfits of crustal rates.

**4.2 Implications of accounting for regional-scale upper structure in future GIA predictions**
Our results show that accounting for regional variability in upper mantle structure has an impact on both sea level and bedrock elevation changes in central West Antarctica over the next few





centuries (Figs. 5, 6). It is well established that the evolution of the West Antarctic Ice Sheet depends on the elevation of the solid Earth and sea level change at the grounding line (e.g., Gomez et al., 2010, 2015, 2024; Konrad et al., 2015; Larour et al., 2019). A sea level feedback on grounding line dynamics arises in the marine basins of West Antarctic Ice Sheet, where solid Earth uplift and sea level fall can act to slow and reduce grounding line retreat (Gomez et al., 2010). Several studies have highlighted the importance of Earth structure when assessing feedbacks between GIA and ice dynamics in the ASE (e.g., Kachuck et al., 2020; Book et al., 2022; Gomez et al, 2024). Recent studies (Gomez et al., 2018, 2024; Coulon et al., 2021) demonstrated the importance of considering lateral heterogeneity in Earth structure at a continental-scale for assessing the impact of the sea level feedback on past and future ice sheet evolution in Antarctica, however, regional-scale variability in upper mantle viscosity has never been accounted for in coupled GIA-ice sheet models.

Across the same TG profile in Fig. 6, Gomez et al. (2024) finds that $O(100 \text{ m})$ of GIA-induced bedrock uplift in a simulation with 3-D, continental-scale viscoelastic Earth model slows and limits grounding line retreat by up to 100 km in 2200 compared to simulations on a rigid bed. Our results indicate that accounting for regional-scale viscosity structure could alter the amount of uplift at the TG grounding line by up to 20% or $O(20 \text{ m})$, which would impact the strength of the sea level feedback in the region. However, it will be necessary to incorporate regional-scale viscosity structure into a coupled model to rigorously quantify the potential impact.

Kachuck et al. (2020) and Book et al. (2022) adopt simplified, 1-D treatments of Earth deformation to explore the sensitivity of ice sheet model projections to Earth structure and find that a laterally homogeneous low viscosity upper mantle (1 x $10^{18}$ – 4 x $10^{18}$ Pa s) reduces projected ice mass loss at PIG and TG, respectively. However, these studies do not account for strong lateral variations in upper mantle viscosity between the grounding line and the interior of the TG and PIG basins (Fig. 1). Our results indicate that this transition from lower to higher viscosity upper mantle material on a transect from the grounding line to the interior of the TG and PIG basins results in higher sea level and lower bedrock elevation predictions at the grounding line of the TG and PIG as soon as 2050 (Figs. 5, 6). As higher sea level at the grounding line will produce greater ice mass flux, the Kachuck et al. (2020) and Book et al. (2022) may over-predict the stabilizing effect of solid Earth



– ice sheet feedback processes at PIG and TG by neglecting lateral variations in upper mantle
viscosity.

Placing high-resolution constraints on the viscoelastic structure of the solid Earth proximal to the modern-day grounding line will be important for improving the accuracy of GIA predictions for ice mass changes projected over the next ~100 years; however, upper mantle viscosity features
located hundreds of kilometers inland may also impact GIA predictions on multi-century timescales (Figs. 5, 6). For example, as discussed in section 3.4, ~10 m lower sea level is predicted in the eastern portion of the TG grounding line in 2300 in the simulation with REG_P due to low viscosity upper mantle material beneath the Byrd Subglacial Basin (Fig. 5h). Our results also suggest that accounting for regional variability in upper mantle viscosity offshore in the Amundsen
Sea may be necessary to accurately constrain solid Earth – sea level feedback processes that likely impacted grounding line retreat from the continental shelf since the Last Glacial Maximum (e.g., Kodama et al., 2023) and grounding line retreat and readvance that has been proposed for the ASE during the Holocene (e.g., Johnson et al., 2022; Balco et al., 2023). Overall, accurately accounting for short-wavelength viscosity features across central West Antarctica, not just proximal to the
modern grounding line, is important for improving constraints on the magnitude of solid Earth – ice sheet feedback processes.

## 5 Conclusions

In this study, we assess the impact of accounting for regional-scale variations in upper mantle
viscosity, at length scales of 50-100 km, on GIA model predictions in central West Antarctica. We show that differences between simulations adopting upper mantle viscosity structure inferred from regional- versus continental-scale seismic imaging are large enough to impact the interpretation of crustal motion observations and reach ~15% of the total predicted sea level change during the instrumental record. Differences of up to ~25% occur between predictions of future sea level
change with continental and regional viscosity models in response to future ice loss out to 2300. The largest discrepancies in sea level change predictions between simulations adopting regional versus continental viscosity models are predominantly found proximal to the grounding line for both modern and projected ice mass changes. Generally, we find that horizontal crustal motion predictions are more sensitive than vertical crustal motion predictions to regional-scale variations



in upper mantle viscosity. While there is considerable mismatch between our model predictions of crustal motion rates and the rates observed at GPS sites in central West Antarctica, our results suggest that it may be important to account for regional-scale viscosity structure in areas with localized high/low viscosity upper mantle features that are not well-resolved in continental-scale viscosity models. Accurately accounting for the transition from lower viscosity upper mantle

material in the coastal ASE to higher viscosities in the interior of the TG and PIG basins will contribute to improving our current understanding of solid Earth – ice sheet – sea level feedbacks in the region.

Our findings highlight that accurately constraining the solid Earth deformational response and sea

level changes associated with modern and future ice mass changes in central West Antarctica will be critical for improving our current understanding of ice sheet stability, the interpretation of geophysical observables, and geological records of ice change. Continuing to improve constraints on 3-D solid Earth structure, both proximal to the modern grounding line and across West Antarctica more broadly, will be important for improving the accuracy of GIA predictions. As up

to 10% faster crustal motion rates are found when considering a 125-year modern ice history compared to a 25-year ice history, further constraints on ice sheet evolution prior to the 1990s will also be important for improving data-model fits of crustal motion rates. Finally, while this study focuses on central West Antarctica, the conclusion that regional-scale mantle structure impacts the predicted spatial pattern and rates of crustal deformation and sea level changes in response to

surface ice loading changes should be considered when modeling GIA in other regions of the planet, especially those underlain by strongly varying solid Earth structure.

**Code and Data Availability:**  The Seakon 3-D GIA model, which has been used in many previous

studies, is described in Latychev et al. (2005a). Efforts are currently underway to make the code publicly    available.    GPS    data    used    for    the    OSU    solution    is    available
[https://www.unavco.org/data/gps-gnss/gps-gnss.html](https://www.unavco.org/data/gps-gnss/gps-gnss.html).

**Author contribution:**  Conceptualization: E.M.L., N.G. Methodology: All. Data acquisition and

curation: T.W. and E.M.L. Investigation: All. Visualization: E.M.L. Funding acquisition: All.



Supervision: N.G. and T.W. Writing – original draft: E.M.L. and N.G. Writing – review and editing: All.

**Competing Interests:** No competing interests to declare.


**Acknowledgements:** This research was supported by Natural Sciences and Engineering Research Council of Canada grant RGPIN-2016-05159 (to N.G.); Canada Research Chairs program grant 241814 (to N.G.). E.M.L acknowledges support from the Wares Postdoctoral Fellowship and the McGill Space Institute Postdoctoral Fellowship. This material is based upon work supported by 805 the National Science Foundation under Award Number 1745074 to the Ohio State University. We acknowledge the work of the OSU ANET team, Demián Gómez, Eric Kendrick, Michael Bevis and David Saddler, for processing the GPS data. We thank Konstantin Latychev for support and advice on working with the Seakon GIA model. We thank the individuals and organizations who have helped with the installation and maintenance of GPS sites across Antarctica. Figures were 810 generated using the Generic Mapping Tools (GMT; Wessel et al., 2019). We thank authors from the Lloyd et al. (2020), Lei et al. (2020), and Afonso et al. (2019) studies for making their models readily available.

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
