# Peer review of "The impact of regional-scale upper mantle heterogeneity on glacial isostatic adjustment in West Antarctica"

_EGUsphere, 2024_

## Author Response (AR1)

We thank the reviewers for their constructive and thoughtful comments, which helped us to improve the manuscript. We have provided our response to reviewer's comments, leaving the original comments in black text and our response in blue text.

We start by noting that both reviewers suggested that the length of the paper may detract from the findings. To reduce the length of the paper, we have moved the extended description covering the process used to construct the regional-scale viscosity model and the comparison between ICE-125 and ICE-25 ice forcings to the Supplementary Material.

**Reviewer 1:**

In this study the authors investigate lateral heterogeneity in upper mantle viscosity below West Antarctica. They compare three different viscosity profiles based on three different seismic models, two that are able to distinguish higher resolution features. They use the viscosity profiles to investigate the effect on GIA in the region using three ice histories: extended modern, modern and future. The results are compared to GPS vertical and horizontal rates. The main findings are that there is up to 20% difference in sea-level change near the grounding line of Thwaites and Pine Island glaciers over the next 300 years. Showing that including higher resolution 3D variations in upper mantle viscosity is important for these crucial and highly dynamic areas.

**General Comment:**

Overall, the paper is interesting, and the science is solid. However, my main concern is that the impact is lost in the sheer length of the paper, with already a lot of figures in the supplementary material. My comments are geared towards simplifying this paper to allow the science to be more impactful.

Thank you for the summary of our study and findings. To shorten the paper and make the science more impactful, we have moved the text and figure detailing the investigation of the 125-year (ICE-125) versus 25-year (ICE-25) ice forcing to the Supplementary Material. We now primarily report on results for simulations adopting ICE-125 in the main text and focus on discussing how different degrees of upper mantle heterogeneity impact GIA model predictions. We have also moved the extended description of the process used to construct the regional-scale viscosity models (REG_P and REG_S) to the Supplementary Material. The text has also been edited and simplified in numerous places in response to specific suggestions and questions below.

**Specific Comments:**

- **Scales**: Abstract line ~18 and introduction. It becomes a bit confusing with the mention of scales here. You mention continental, regional, local scale, and then >150km, 50-100km, and 50-200km separately. It would be useful to clearly define the spatial scale in km of each of the continental, regional and local scale – perhaps lines on 78 and 80. Furthermore, "local scale" is then not mentioned in the results – they are either CONT or REG. Is the regional scale actually representative of regional and local combined? Please clarify and simplify. Line 86, again clarify the scale in km. We have added in an

approximate length-scale for regional-scale seismic imaging to Section 1: "…recent regional-scale (~400 - 1000 km length-scale) seismic imaging…". We originally mentioned local-scale imaging in reference to the Lucas et al. (2021) study, which images uppermost mantle structure near the grounding lines of Thwaites and Pine Island glaciers; however, this could also be considered regional-scale imaging. Therefore, we have simplified this by removing the reference to local-scale imaging here and throughout the manuscript.

- **Section 2.1**: I think this section is too long. We have shortened this section by moving the description of the process used to construct the regional-scale viscosity models (REG_P and REG_S) to the Supplementary Material. Additionally, the section describing "Regional upper mantle viscosity features" has been moved to the Results section (now Section 3.1).

  - 1D model – This is introduced as if there will be lots of comparisons between the 1D and 3D, but I can only see it in Figure 6 of the main paper. This is fine, and I'm more interested in the differences between the 3D models anyway, but perhaps state on line 132 that most 1D results are in supp info. First, we have changed the name of the "1D" model to "1D_WAIS". We have added that "As this study primarily focuses on GIA predictions using 3-D viscosity models, results from simulations with 1D_WAIS are largely presented in the Supplementary Material (Fig. S4-S7)."

  - Line 162: why is this 1D viscosity profile different to that mentioned earlier? The 1-D reference viscosity profile that we describe here is distinct from the 1D_WAIS viscosity model (previously just "1D") described earlier. As described in the preceding text, this is a reference profile on which "Lateral mantle viscosity variations are superimposed" for the 3-D viscosity models. It is a global average? If so, I would suggest to delete the end of the sentence "consistent with previous GIA modelling…. In west Antarctica" as this makes it sound like previous modelling shows UM viscosity in West Ant to be 5x10^20. You are correct, this reference viscosity profile is representative of a global average. We have removed the portion of the sentence referring to West Antarctica for clarity.

  - Line 175 to 181: Suggestion - I think you can simplify this to: "…47degS (Fig 1a). [delete the next bit] Global viscosity variations are estimated from GLAD because it offers improved structure compared to S326ANI that was used in Gomez 2024" We have simplified the sentence based on your suggestion.

  - Paragraphs starting line 217 and 234: I think most of this detail should be moved to supp info, particularly (all of?) the second paragraph. We have kept key details about the method followed to merge the regional- and continental-scale models

in the main text and moved much of the extended detail on steps taken to merge the models and scale to Section S1 of the Supplement.

○ I'm interested to know how you join the regional models into the ANT20 model – in terms of the edge effects. Do you have to do smoothing where these two datasets are joined together? Or are they referenced to the same background model so its not an issue? Perhaps include in the supp information with (d). We have included additional details clarifying  the methods used to merge the models in Section S1. We adopt a ~75 km smoothing band along the edges of the regional and continental models to ensure a gradual transition between the model. These two models are not referenced to the same background model because the regional seismic models provide anomalies relative to an unknown background model mean.

The added text reads: "When inserting the regional seismic models into ANT-20, a ~75 km smoothing band is adopted along the edges of the regional and continental models to ensure a gradual transition between the models (Fig. 1). The relative travel-time tomography approach adopted by Lucas et al. (2020) provides velocity anomalies relative to an unknown background mean rather than absolute velocities. In contrast, mantle velocity anomalies in ANT-20 are reported relative to the 1-D Earth model STW105 (Kustowski et al., 2008). Consequently, a 0% velocity anomaly in the Lucas et al. (2020) regional seismic models does not correspond to a 0% velocity anomaly in the ANT-20 model. To ensure consistency amongst the regional-scale viscosity models and the CONT viscosity model, we use the maximum and minimum viscosity bounds from the CONT viscosity model as a guide for constructing the regional viscosity models, ensuring upper mantle viscosities in the regional models remain within these viscosity bounds for central West Antarctica."

○ Line 294: "error" feels a bit harsh. Change to limitations? Maxwell is still valid! Could you return to this in the discussion – any idea what might the impact be if the experiments were repeated with a transient viscosity? Good point – we changed "error" to "limitation" here.

Because the manuscript is already long, we do not explore the impact of adopting transient viscosity. To return to the idea that transient or non-linear rheology may need to be considered, we have added a sentence to the discussion noting that "…discrepancies between predicted and observed crustal rates in the ASE may indicate that mantle deformation is better represented by transient or non-linear rheology (e.g., Lau et al., 2021; Blank et al., 2021; Ivins et al., 2022) rather than the adopted Maxwell viscoelastic rheology."

• **Section 2.2:**

- o Just on terminology here – modern/extended modern ICE-25/ICE-125. Later on ICE-125 is also referred to as "modern ice mass change" e.g. section 3.2 title. Either change "modern" to "extended modern" where referring to ICE-125, or state that there are two version of modern in this section, and then be clear which modern you're talking about. Thank you for catching the inconsistency in our terminology. As we have moved the discussion comparing results in simulations using ICE-25 and ICE-125 to the supplement, we have updated the text to refer to ICE-125 as the modern ice loading model.

- o Figure 2 caption – some of this description is repeating what is in the main text on how the ice histories are constructed. Not needed in the caption. Panel b – over what area or drainage basic is this calculated? We have removed the description of how ice histories are constructed from the caption. We also specify that cumulative mass is calculated for the entire Antarctic Ice Sheet.

- **Section 4.1**: comparison to horizontal rates. I question the inclusion of comparing to horizontal GPS rates (line 656 to 685; Figure 7b and c). As the authors state, the horizontal GPS are not corrected for any kind of background GIA rate, and so already the comparison between the model and GPS observed rates in 7b is not going to match. I think this whole section could be removed or put into supplementary information. GIA studies in Antarctica (and elsewhere) often overlook the comparison between predicted and observed horizontal crustal rates. However, horizontal crustal rates undoubtedly provide valuable insight into both Earth structure and ice history. To underscore the importance of improving data-model misfits for horizontal crustal rates, we prefer to leave the discussion of horizontal crustal rates in the main text. While we acknowledge that our original manuscript was too long, we feel that the other edits we have made to address this will suffice.

**Minor Comments:**
Line 51: Change to "Accurate modelled predictions" since empirical estimates of GIA do not rely on Earth properties. Added in modeled to "accurate predictions".

Line 111: remove "the" > according to Wan. We think that it reads better with the "the" before Wan.

Line 175: put the transition zone depth in brackets here to help the reader, and also for the Moho on line 216. Replaced with 670-km depth for clarity. We cannot reference a single depth for the Moho here because it is variable.

Line 194: something has happened with the end of this sentence – "how to correct". We have added on the rest of the sentence. It was meant to say "how to correct for underestimated seismic velocity anomaly amplitudes in the Lucas et al. (2020) models."

Figure S1: the scale of 200km looks like a label, a bit confusing as I thought I was looking at two slices at 200km depth. We have changed the scale to show 400 km instead of 200 km to reduce confusion.

Line 388: modern or extended modern? As we have moved discussion of simulations adopting ICE-25 versus ICE-125 to the supplement, we are now referring to simulations with ICE-125 as modern throughout.

Line 416: Super picky – but can you change the reference to Fig 3a where PSK is also labelled? To save a lot of going backwards and forwards to different figures. Yes - we have changed the reference to Fig. 3a instead.

Lines 541-551 and the next paragraph: I'm not sure I follow here. Does "higher relative sea level" mean "less sea level fall"? So the CONT model predicts more sea level fall than the REG models? Might be worth just clarifying here and relating to the colours on the graph. Yes, higher relative sea level does mean less sea level fall. We have changed several references to "higher relative level" to "less sea level fall" to make the two paragraphs read more clearly.

Figure 6: Is this figure really needed? Yes, we think that this figure (now Fig. 5) is useful for visualizing differences in grounding line elevation amongst the simulations in a different way than what is shown in Fig. 4. We have added some further references to this figure in the text so that it is now better utilized to explain our findings.

Figure 7: The symbols are hard to see because they are so small. In fact, the statement on line ~650 "introducing regional variations in viscosity reduces differences between predicted and observed crustal rates at a number of sites" is really hard to see in this figure. It looks like the opposite is true. I wonder if REG_P ICE-25 should be removed, REG_S colour could be changed to group it in with REG_P, leaving CONT and 1D in blue/green. We have updated Fig. 6 (previously Fig. 7) to offset the symbols from each other horizontally for each GPS station. We have removed predictions from the REG_P ICE-25 simulation in Fig. 6 and moved them to a new figure in the supplement (Fig. S8). Fig. S8 includes predictions from simulations adopting the ICE-25 and ICE-125 ice models with the REG_P viscosity model for comparison. It would not make sense to make the symbols from REG_P and REG_S the same color as they are predictions from two different simulations.

Section 4.1: Is there also an elastic correction made to the GPS rates for SMB? No, an elastic correction is not made to the GPS rates for SMB. We have added "An elastic correction for surface mass balance is not applied to the GPS crustal motion rates. " to Section 2.3.

Line 716: What is this "O" in O(100m)? Please define. The italicized O can be used to denote "on the order of". To reduce confusion, we have removed the $O$ and replaced it with "on the order of".

Line 720: alter the amount of uplift by 20% impacting the strength of the feedback…. Which way does this go – does it increase or decrease the strength of the feedback, i.e., is it stabilising or destabilising/less stabilising by including regional scale 3D variations in viscosity? We further clarify that "accounting for regional-scale viscosity structure could *reduce* the amount of uplift at the TG grounding line by up to 20% (or up to 20 m), which would *negatively* impact the strength of the sea level feedback in the region."

Code availability: It's great to see that this code is on its way to being made publicly available.
* * *
**Reviewer 2:**

**General Comments:**

The submitted manuscript investigates the impact of regional-scale (50-100 km) lateral variations in mantle viscosity on GIA model predictions in Antarctica. Understanding heterogeneity in mantle viscosity is important for interpreting geophysical data and modeling ice sheet-solid Earth feedbacks, and has critical implications for the future of the Antarctic Ice Sheet.

The authors employ two previously published regional tomography models for their work. They stitch these models into continental- and global-scale topography models and convert the velocity anomaly to a viscosity anomaly. Their final models show similar features to previously published work, however, they highlight shorter wavelength variability, particularly in the Amundsen Sea.

Model predictions of solid Earth deformation, gravitational potential change, and relative sea-level change with the regional viscosity structure show significant differences from the continental and 1D models. The authors highlight the importance of these findings by comparing their deformation estimates with GPS data from across West Antarctica. While they do not find that the regional model improves the overall fit to the data, they show convincingly that these differences are significant and warrant further exploration.

This work presents a specific and important scientific problem and investigates it with sound methods. I find the work to be robust and believe it will make a solid contribution to our understanding of GIA in Antarctica. That said, I have outlined some issues below that should be addressed.

Thank you for the summary of our study and findings.

**Major Points:**
In isolation, the parameter choices for the 1D viscosity model are well justified, as is the justification for the 1-D reference profile from which 3D anomalies are calculated. However, I find it confusing to use two different 1D profiles in the same study. It would be much easier to

interpret the differences between the 1D and CONT/REG_P/REG_S models if those 3D models used the same reference 1D case. As written, it is unclear to what degree differences between 1D and 3D are due to the actual anomalies in the 3D model or due to the difference in the mean viscosity value (approximately an order of magnitude).

Adopting the same viscosity profile for the 1-D reference profile as used in the 1D_WAIS viscosity model (previously named "1D") would result in unrealistically low viscosity values in regions of low viscosity within the 3-D models. Additionally, we would not be able to capture near-average and high upper mantle viscosities found in other regions of West Antarctica by adopting a lower-viscosity 1-D reference profile. For these reasons, we argue that it is better to retain the current 1-D reference profile used in the 3-D models.

The adopted 1D_WAIS viscosity model we use facilitates the comparison of model predictions from this study with previous work aimed at understanding solid Earth – ice sheet feedbacks locally in the Thwaites and Pine Island glacier regions (e.g., Kachuck et al., 2020; Book et al., 2022). We have added text to Section 2.1 to further explain the rationale behind adopting the 1D_WAIS profile, including "..we adopt one 1-D (i.e., radially varying) Earth model representative of the structure of low viscosity zones in West Antarctica inferred in the literature.." and "The adopted 1D_WAIS model allows for more direct comparison with recent studies on solid Earth – ice sheet feedbacks in the ASE, which use 1-D Earth models with upper mantle viscosities in the $10^{18}$-$10^{19}$ Pa s range (e.g., Kachuck et al., 2020; Book et al., 2022)."

I think the issue of vertical smearing in body wave tomography identified in Lucas et al. (2020) is overlooked. The authors explain that the checkerboard tests determine the amplitude recovery values, but do not explain how this is related to vertical smearing of the velocity anomaly. This limits the vertical resolution of their REG_X models and certainly has an impact on the GIA results they obtain so it should be discussed. It might also be noted that their vertical resolution is quite different to the ANT-20 model.

We do acknowledge that "vertical resolution is limited in the Lucas et al. (2020) regional seismic models; however, resolution tests indicate that the imaged velocity anomalies primarily originate from mantle structure between the Moho and ~250 km depth." in Section 2.1.2. Additionally, we note that vertical smoothing in ANT-20 is fixed at ~45 km for all depths in Section 2.1.1. Per the suggestion to expand upon vertical smearing in the body wave tomography, we have added in an additional sentence to Section 2.1.2 stating: "Given that resolution tests show >150 km of vertical smearing in the Lucas et al. (2020) regional models, ANT-20 likely provides superior resolution of vertical variability in upper mantle structure with vertical smoothing fixed at ~45 km."

It is unclear to me how the regional models are inserted into the ANT-20 model. The relative travel time models should only provide velocity perturbations, while the ANT-20 adjoint model provides absolute velocities. It seems that to insert the regional model would require making some correction based on the ANT-20 mean over the same spatial domain. Could the authors please explain their methods and reasoning here?

We have added additional details to explain how the regional models are inserted into the ANT-20 model in Section S1 of the Supplement.

The added text reads: "The relative travel-time tomography approach adopted by Lucas et al. (2020) provides velocity anomalies relative to an unknown background mean rather than absolute velocities. In contrast, mantle velocity anomalies in ANT-20 are reported relative to the 1-D Earth model STW105 (Kustowski et al., 2008). Consequently, a 0% velocity anomaly in the Lucas et al. (2020) regional seismic models does not correspond to a 0% velocity anomaly in the ANT-20 model. To ensure consistency amongst the regional-scale viscosity models and the CONT viscosity model, we use the maximum and minimum viscosity bounds from the CONT viscosity model as a guide for constructing the regional viscosity models, ensuring upper mantle viscosities in the regional models remain within these viscosity bounds for central West Antarctica."

The investigation of the 125 versus 25 year ice forcings is very thorough, however, since this paper is focused on the spatial pattern of solid Earth deformation rather than the ice reconstructions, I think including a lengthy discussion of both of these in the main text is unnecessary and detracts from the most important findings of the study. The main takeaway from Figure 4 is that there is more deformation in the region of load change in the longer loading scenario, which is not surprising and does not add much additional information in terms of how regional- versus continental-scale viscosity models behave. I suggest moving this figure to the supplement and focusing mainly on the 125 yr history in the main text.

We agree with you that the assessment of GIA predictions using 125- versus 25-year ice histories detracts from the focus of the paper and have moved the text and figure comparing results using 125-year versus 25-year ice histories to the supplement (now S1 Comparison of GIA model predictions for simulations adopting ICE-125 versus ICE-25). Fig S7 in the supplement originally showed the difference in model predictions in simulations using ICE-125 versus ICE-25 for the REG_S, CONT, and 1D viscosity models, so we have combined the plots from Fig. 4 with those in Fig. S7 for simplicity.

I suggest moving the discussion of specific features in the Earth models (section 2.1.3) to the results section. These features are a product of the conversion from velocity to viscosity and thus belong in the results. I think this will also improve the readability to have these features highlighted closer to where they are discussed in detail at the end of the paper. A sentence in the beginning of the results section (lines 379-380) actually indicates that this was the intention, but for some reason it was placed in the methods.

We agree that moving the section on "Regional upper mantle viscosity features" to the results section improves the readability of the paper. Section 2.1.3 is now included in the results section as Section 3.1, with some minor edits.

Ideally, the authors would address the issue of future projections of GIA by running a fully coupled simulation with a dynamic ice model. This would be the only way to fully understand the impact that their REG_X viscosity models might have on groundline dynamics and GIA. Without coupled simulations, it is hard to interpret their results since these ice-loading models (ICE-FUT) are based on different viscosity structures. At a minimum, it would be helpful to plot the groundline evolution (as calculated by the floatation criterion in the Seakon) in Figure 6 for

different models to assess the potential impact this viscosity structure might have on ice stability.

We agree that coupled simulations with a dynamic ice model would be the best way to understand the impact of incorporating regional upper mantle structure on grounding line dynamics and, motivated by the results of this investigation, we foresee pursuing such an investigation in the future. Such simulations are highly computationally expensive and as illustrated in Gomez et al. (2024) with the continental viscosity model, the strength and nature of the feedback is sensitive to the climate forcing. We thus feel that a thorough exploration merits its own study.

As the flotation criterion in Seakon does not accurately capture the feedbacks between GIA and ice sheet dynamics, we feel that it would be misleading to show and challenging to interpret grounding line positions calculated using the floatation criterion for each viscosity model. Thus, we only show grounding line positions predicted using the ICE-FUT model in Figs. 4-5.

Comparing predicted and observed crustal rates is a nice way to highlight the importance of regional viscosity models. However, it is unclear to what degree either model performs better/worse than the 1D or CONT models in matching observed vertical rates overall. For each model, I would suggest reporting an average residual between predicted and observed rates (for the vertical rates at least). This would provide context for the authors' argument that regional models are necessary to accurately interpret the data (lines 687-690).

To make the performance of each viscosity model more clear, we have added text to Section 4.1 in which we report the average residuals between the observed and predicted vertical crustal rates for the 1D, CONT, REG_P, and REG_S models.

The added text reads: "Across all GPS sites, the average residual between observed vertical crustal rates (corrected using Gomez et al. (2018) model predictions) and model predictions is 9.1 mm/year for simulations adopting the 1D_WAIS viscosity model. In comparison, the average residuals for simulations using the CONT, REG_P, and REG_S models are 6.1 mm/year, 6.7 mm/year, and 7.2 mm/year, respectively."

In keeping with the central theme of the paper to investigate the impact of shorter wavelength features, I think it would be useful to have a paragraph in the discussion about whether the resolution of the adopted models are good enough. Should the GIA community strive for even higher resolution? What resolution is unnecessarily high? Is there evidence to suggest low/high viscosity zones may exist that these new models do not capture? I think the authors have valuable insight to contribute and could strengthen the overall impact of the paper by addressing these questions.

We have expanded Section 4.2 to discuss how continuing to improve constraints on various low viscosity mantle features across West Antarctica will help to improve the accuracy of GIA predictions and likely reduce data-model misfits. More specifically, we expand upon the discussion of how constraining the geometry of localized low viscosity mantle features, like that found beneath the Byrd Subglacial Basin, will help improve GIA predictions:

"For instance, as discussed in Section 3.4, the simulation with REG_P predicts ~10 m lower sea level along the eastern portion of the TG grounding line in 2300 due to the presence of low viscosity upper mantle material beneath the Byrd Subglacial Basin (Feature C; Fig. 4b, h), a graben that likely underwent Neogene extension (e.g., LeMasurier et al., 2008; Granot et al., 2010; Lucas et al., 2020). The influence of accounting for such a localized upper mantle feature on GIA predictions underscores the need for improved geophysical constraints on the spatial distribution and geometry of similar low viscosity mantle features across West Antarctica. In particular, refining constraints on Earth structure in other areas that may have experienced localized Neogene extension – such as the Pine Island Rift (beneath Pine Island Glacier) and Bentley Subglacial Trench (adjacent to Byrd Subglacial Basin) – as well as various Cenozoic volcanic provinces will improve the accuracy of GIA predictions and reduce data-model misfits."

Minor Points:

Line 80: Could you provide approximate length scales of 'local' and 'regional'? Reviewer 1 also commented on this, and we have added in approximate length scale for regional-scale imaging and removed the reference to 'local' scale imaging for clarity. We originally mentioned local-scale imaging in reference to the Lucas et al. (2021) study, which images uppermost mantle structure near the grounding lines of Thwaites and Pine Island glaciers; however, this could be considered regional-scale imaging. Therefore, we have simplified this by removing the reference to local-scale imaging.

Line 87: Similarly it would be nice to define clearly what is exactly meant by regional and how much it differs from continental. We have further specified the scale of regional-scale imaging in the introduction and note that regional-scale imaging has revealed heterogeneity at the glacial-basin scale: "glacial-basin scale investigations of GIA have remained elusive due to limited seismic resolution. However, benefiting from improved seismic station coverage in West Antarctica, recent regional-scale (~400 - 1000 km length-scale) seismic imaging has revealed notable heterogeneity in upper mantle seismic velocities within the TG and PIG glacial drainage basins…"

Line 88: It would be useful to define 'relative sea level' here or somewhere in the introduction or at the beginning of the results section. This term can be confusing especially in studies like this where simulations are run both from the past to present and from present into the future. We now define relative sea level in the last paragraph of Section 1 – "…we evaluate the impact of regional-scale variability in upper mantle viscosity on predictions of changes in relative sea level (i.e. the height of the sea surface equipotential relative to the solid surface)".

Line 194: Looks like something happened to part of a sentence here. You are correct, and we have added on the rest of the sentence. It was meant to say "how to correct for underestimated seismic velocity anomaly amplitudes in the Lucas et al. (2020) models."

Line 405: I would suggest changing "higher relative sea level" to "less relative sea level fall". It may be confusing to some who are less familiar with GIA and thinking about 'relative' sea level

to interpret this sentence. Saying "less sea level fall" more directly gets to the point that over the length of the simulation there is less change in RSL in the 1D model. (On a very technical level, the statement that RSL is 'higher' at present is also confusing since the final prediction of the Seakon (or any) GIA code in a historical/paleo simulation is that RSL=0.) We have changed the wording from "higher relative sea level" to "less sea level fall". We agree with you that this phrasing will reduce confusion for readers who are less familiar with GIA.

I might also change 'lower magnitude vertical crystal rates' to 'lower magnitude modern-day vertical crustal rates' if that is what is plotted. We have changed the wording to "lower magnitude modern-day vertical crustal rates".

Line 416: Could you label the PSK region in Figure 3? The PSK region is labeled in Fig. 3a, so we have changed it to reference Fig. 3a instead of Fig. 1d.

Figure 3: The label on the left for plots (a) (b) and (c), should more accurately be 'change in relative sea level' or 'relative sea level at 125 ya'. We have changed the label on the left for plots (a), (b), (c) to "relative sea level change". We have also updated several plot labels in the supplement from "relative sea level" to "relative sea level change".

Line 419-420: Following the comment about line 405, I would change the language to "less change in relative sea level". Changed the wording to "less change in relative sea level".

Figure 4: It would be useful to readers to state which model is subtracted from the other in the figure caption. (Same with Figure S4). We have added to the second sentence of the caption that we do "(REG_P minus CONT predictions)" and "(REG_S minus CONT predictions)" To further clarify for readers. We have also added "(1D predictions minus CONT predictions)" to the caption of Fig. S4.

Figure 5: I would add in the caption a line to aid the reader in interpreting the REG_P - CONT and REG_S - CONT plots. Something like: "positive values indicate overall less sea level fall in the regional model during the labeled time period". We have added "In the REG_P – CONT and REG_S – CONT plots, positive values indicate overall less sea level fall in simulations adopting the regional model during the labeled time period, while negative values indicate greater overall sea level fall." to the caption of Fig. 4 (old Fig. 5). Additionally, we have added the line "In (b-c), positive values correspond to less sea level fall predicted in simulations adopting the regional viscosity models compared to those adopting CONT, while negative values correspond to greater sea level fall." to the caption of Fig. 3 to aid reader interpretation.

Line 543-544: Here, I think it makes sense to say 'higher relative sea level is predicted' since RSL can vary between different models in the future. But I would clarify this in the sentence and also add a point about what this means for overall sea level change to aid the reader. I would correct this by changing:

"Compared to simulation with CONT, higher relative sea level (+1.31 m compared to CONT) is predicted in the central PIG basin with the REG_P model"

To something like:

"In the central PIG basin, the REG_P predicts overall less sea level fall from 1950 to 2050 compared to CONT, resulting in 1.31 m higher relative sea level in 2050"
Thank you for the suggestion on how to make the wording in this sentence clearer. We have updated the sentence based on your suggestion.

Line 543-548: I found this paragraph confusing. I would suggest revising to get at the really intriguing differences between the REG_P at 2050 (which predicts less sea level fall overall) and the REG_S (which has a northern region with more sea-level fall and a southern region near Thwaites with less). We have done significant revisions to this paragraph to make it less confusing. It now reads:

"In the central PIG basin, the REG_P simulation predicts less overall sea level fall from 1950 to 2050 compared to the CONT simulation, which ultimately produces 1.31 m higher relative sea level in 2050 in the REG_P simulation (Fig. 4a-b). Unlike the REG_P simulation, greater overall sea level fall (-0.49 m) is predicted from 1950 to 2050 in the northern PIG basin in the simulation adopting REG_S versus CONT (Fig. 4a, c). These discrepancies in relative sea level predictions in the PIG basin can be attributed to differences in the REG_P and REG_S viscosity models. More specifically, the presence of low-viscosity Feature B in REG_S, which is not as prominent in REG_P, is what produces greater overall sea level fall in the REG_S simulation."

Figure 6: In panels A and B, could you also plot the viscosity anomalies along the profile for REG_P and REG_S? We find it difficult to effectively visualize the bedrock elevation profiles and changes in bedrock elevation if the viscosity anomalies are also plotted in Fig. 5 (previously Fig. 6); therefore, we have left the figure as is.

Figure 7: The symbols in the A and B are hard to read. I would suggest offsetting the symbols horizontally by a small amount and adding dashing lines to separate each station. REG_P (ICE-25) could also be in the supplement. We have taken your suggestion to offset the symbols in Fig. 6 (old Fig. 7) and have added dashed lines to separate each station. We have removed predictions from the REG_P ICE-25 simulation and moved them to a new figure in the supplement (Fig. S8). Fig. S8 includes predictions from simulations adopting the ICE-25 and ICE-125 ice models with the REG_P viscosity model for comparison.

Line 720: Could you say in which direction it would alter it? More or less? We further clarify that "accounting for regional-scale viscosity structure could reduce the amount of uplift at the TG grounding line by up to 20% (or up to 20 m), which would negatively impact the strength of the sea level feedback in the region."

---

## Referee Report (RR1)

I appreciate the authors addressing each comment carefully. I believe the revised manuscript is much improved and will make an important contribution to The Cryosphere. I'd only ask that the reviewers address one issue detailed below. The original comment is in black, their response in blue, and my final comment in red.

Ideally, the authors would address the issue of future projections of GIA by running a fully coupled simulation with a dynamic ice model. This would be the only way to fully understand the impact that their REG_X viscosity models might have on groundline dynamics and GIA. Without coupled simulations, it is hard to interpret their results since these ice-loading models (ICE-FUT) are based on different viscosity structures. At a minimum, it would be helpful to plot the groundline evolution (as calculated by the floatation criterion in the Seakon) in Figure 6 for different models to assess the potential impact this viscosity structure might have on ice stability.

We agree that coupled simulations with a dynamic ice model would be the best way to understand the impact of incorporating regional upper mantle structure on grounding line dynamics and, motivated by the results of this investigation, we foresee pursuing such an investigation in the future. Such simulations are highly computationally expensive and as illustrated in Gomez et al. (2024) with the continental viscosity model, the strength and nature of the feedback is sensitive to the climate forcing. We thus feel that a thorough exploration merits its own study.
As the flotation criterion in Seakon does not accurately capture the feedbacks between GIA and ice sheet dynamics, we feel that it would be misleading to show and challenging to interpret grounding line positions calculated using the floatation criterion for each viscosity model. Thus, we only show grounding line positions predicted using the ICE-FUT model in Figs. 4-5.

I agree that Seakon does not accurately capture GIA-ice sheet feedbacks. However, the purpose of Figure 8 is to demonstrate the potential impact that GIA could have on ice sheet dynamics. This impact is mostly related to where the ice is grounded, so it seems important to show whether the Seakon model predicts large changes in where the ice sheet is floating. This would provide a better understanding of the impact of regional viscosity structure on ice sheet dynamics than simply the amount of uplift along a transect.

---

## Author Response (AR2)

I appreciate the authors addressing each comment carefully. I believe the revised manuscript is much improved and will make an important contribution to The Cryosphere. I'd only ask that the reviewers address one issue detailed below. The original comment is in black, their response in blue, and my final comment in red.

Ideally, the authors would address the issue of future projections of GIA by running a fully coupled simulation with a dynamic ice model. This would be the only way to fully understand the impact that their REG_X viscosity models might have on groundline dynamics and GIA. Without coupled simulations, it is hard to interpret their results since these ice-loading models (ICE-FUT) are based on different viscosity structures. At a minimum, it would be helpful to plot the groundline evolution (as calculated by the floatation criterion in the Seakon) in Figure 6 for different models to assess the potential impact this viscosity structure might have on ice stability.

We agree that coupled simulations with a dynamic ice model would be the best way to understand the impact of incorporating regional upper mantle structure on grounding line dynamics and, motivated by the results of this investigation, we foresee pursuing such an investigation in the future. Such simulations are highly computationally expensive and as illustrated in Gomez et al. (2024) with the continental viscosity model, the strength and nature of the feedback is sensitive to the climate forcing. We thus feel that a thorough exploration merits its own study.
As the flotation criterion in Seakon does not accurately capture the feedbacks between GIA and ice sheet dynamics, we feel that it would be misleading to show and challenging to interpret grounding line positions calculated using the floatation criterion for each viscosity model. Thus, we only show grounding line positions predicted using the ICE-FUT model in Figs. 4-5.

I agree that Seakon does not accurately capture GIA-ice sheet feedbacks. However, the purpose of Figure 8 is to demonstrate the potential impact that GIA could have on ice sheet dynamics. This impact is mostly related to where the ice is grounded, so it seems important to show whether the Seakon model predicts large changes in where the ice sheet is floating. This would provide a better understanding of the impact of regional

First, we would like to thank the Reviewer for their feedback and continued interest in improving this manuscript.

We have now plotted the projected grounding line position for the REG_P simulation in dark gray dashed lines in panels c-h of Figure 5 (which is the figure that the Reviewer refers to as Figure 8 above). In the caption, we specify that the "..grounding line positions are calculated by applying the floatation criterion to the output of the REG_P simulation". We also note that "Because projected grounding line positions from simulations with CONT, REG_S, and 1D_WAIS show minimal difference from REG_P, only grounding line positions from the REG_P simulation are plotted" in the caption. We

believe that the changes made to Figure 5 now illustrates the amount of change that Seakon predicts in where the ice sheet is floating versus that prescribed by the ICE-FUT model.

---

## Author Response (AR3)

Dear Dr. De Rydt,

We are now in the process of making all the Seakon model output used in this study available on the Federated Research Data Repository (FRDR). It will be available at https://doi.org/10.20383/103.01249 once the files have been reviewed by the Data Repository managers. We have added information to the "Code and Data Availability" section on how to access the model output.

We would like to thank you for your work as an editor on this manuscript.

Sincerely,
Erica Lucas, Natalya Gomez, Terry Wilson